# Better turbulence spectra from VAD scanning wind lidar

**Felix Kelberlau[1] and Jakob Mann[2]**

[1]NTNU, Department of Energy and Process Engineering, Norwegian University of Science and Technology, 7491 Trondheim, Norway
[2]DTU Wind Energy, Technical University of Denmark, 4000 Roskilde, Denmark

**Correspondence:** Felix Kelberlau (felix.kelberlau@ntnu.no), Jakob Mann (jmsq@dtu.dk)

**Abstract.** Turbulent velocity spectra derived from velocity-azimuth display (VAD) scanning wind lidars deviate from spectra derived from one point measurements due to averaging effects and cross-contamination among the velocity components. This work presents two novel methods for minimizing these effects through advanced raw data processing. The squeezing method is based on the assumption of frozen turbulence and introduces a time delay into the raw data processing in order to reduce cross-contamination. The 2-beam method uses only certain laser beams in the reconstruction of wind vector components to overcome averaging along the measurement circle. Models are developed for conventional VAD scanning and for both new data processing methods to predict the spectra and identify systematic differences between the methods. Numerical modeling and comparison with measurement data were both used to assess the performance of the methods. We found that the squeezing method reduces cross-contamination by eliminating the resonance effect caused by the longitudinal separation of measurement points, and also considerably reduces the averaging along the measurement circle. The 2-beam method eliminates this averaging effect completely. The combined use of the squeezing and 2-beam methods substantially improves the ability of VAD scanning wind lidars to measure in-wind ($u$) and vertical ($w$) fluctuations.

## 1 Introduction

Wind speed measurements are an integral element of wind site assessment. Traditionally such measurements have been based on in situ sampling with anemometers attached to tall meteorological masts that reach up to hub height. Such masts are immobile and expensive to erect. It is therefore favorable to implement remote sensing devices, such as conically scanning profiling lidars, that measure wind velocities at adjustable height levels above the ground remotely.

Pulsed and continuous-wave wind lidars are the two types of profiling lidars that are currently commercially available. The velocity-azimuth display (VAD) scanning strategy was introduced by Browning and Wexler (1968) and is usually applied for continuous-wave profiling lidars like the ZX 300 (previously ZephIR 300) produced by Zephir Ltd. / ZX Lidars. Advanced processing of VAD acquired data is the object of investigation here.

Validation studies that compare measurements from meteorological masts and ground based profiling lidars report good agreement for first order statistics, namely the 10-minute mean wind velocities and directions (Kindler et al., 2007; Smith et al., 2006; Medley et al., 2015; Kim et al., 2016). The estimation of second-order statistics of the turbulence in the wind by means of VAD scanning pulsed Doppler lidar was first demonstrated by Eberhard et al. (1989). But such turbulence estimates from VAD scanning lidars deviate from classical measurements with cup or sonic anemometers (Sathe and Mann, 2013; Peña et al., 2009; Canadillas et al., 2010). Sathe et al. (2011) model the second order statistics of pulsed and continuous-wave profiling lidars. The resulting velocity variances are influenced by the effects that arise from sensing the 3-dimensional wind field by averaging over spatially distributed volumes. In order to better understand the actual behavior of the lidar in comparison to reference measurements, turbulence spectra of the three wind components $u$, $v$ and $w$ can provide much-needed insight. Sathe and Mann (2012) model and analyze turbulence spectra, but only for pulsed lidars that use Doppler beam swing (DBS) scanning. A simplified model for turbulence spectra from VAD scanning wind lidars is presented in Wagner et al. (2009). However, it does not include the effect of cross-

contamination and cannot be used to predict the turbulence spectra of real lidars.

The six-beam method developed by Sathe et al. (2015) is an alternative to VAD scanning that results in more accurate 5 second order statistics of turbulence. But its application requires a vertical laser beam and a half cone opening angle of 45°, which makes it not usable with commercially available profiling wind lidars.

Newman et al. (2016) propose another method to com-10 pensate for the contamination by means of autocorrelation functions derived from collocated mast measurements. This method is, however, only applicable when a meteorological mast is available. In comparing and evaluating the ability of different lidar scanning strategies to measure turbulence, 15 Newman et al. (2016) concludes that cross-contamination of the different velocity components is one of the primary disadvantages of current profiling lidars.

The research presented here demonstrates two methods aimed at overcoming the effects of cross-contamination and 20 averaging along the measurement circle that are inherent in the standard VAD scanning strategy. Both methods are based on modified line-of-sight velocity data processing and can be applied to currently available lidars without changes in their hardware. The line-of-sight averaging effect remains unre-25 solved.

The first method incorporates Taylor's frozen turbulence hypothesis and introduces a time lag into the wind vector reconstruction process. Bardal and Sætran (2016) measure two-point correlations of horizontal wind speeds from two 30 meteorological masts that are separated by $79\,\mathrm{m}$ in line with the mean wind direction. They find that the cross-correlation coefficient is around 0.8 when a temporal lag compensates for the time required for the wind to cover the distance between the two measurement points. Without delaying the sig-35 nal, the cross-correlation coefficient reaches only half of that value. Applied to VAD scanning lidars, that justifies the assumption that when the processing of line-of-sight measurement data is delayed by the time needed to cross the measurement circle, the lidar measurements will be more real-40 istic. This approach is hereafter called "squeezing" and reduces the cross-contamination effect that currently distorts the shape of turbulence spectra acquired with VAD scanning lidars.

The second method is to use only the radial velocities from 45 lines-of-sight that point into the mean wind direction (downwind) and against it (upwind) to determine the components of the wind that are oriented in-line with the mean wind direction ($u$) and vertical ($w$). This eliminates the averaging along the measurement circle.

50 The aim of the research presented here is to demonstrate whether one of the two modified data processing algorithms or their combination leads to improved turbulence measurements from standard VAD wind lidars. Both methods are modelled and applied on the same measurements individu-55 ally, and their effects are discussed separately.

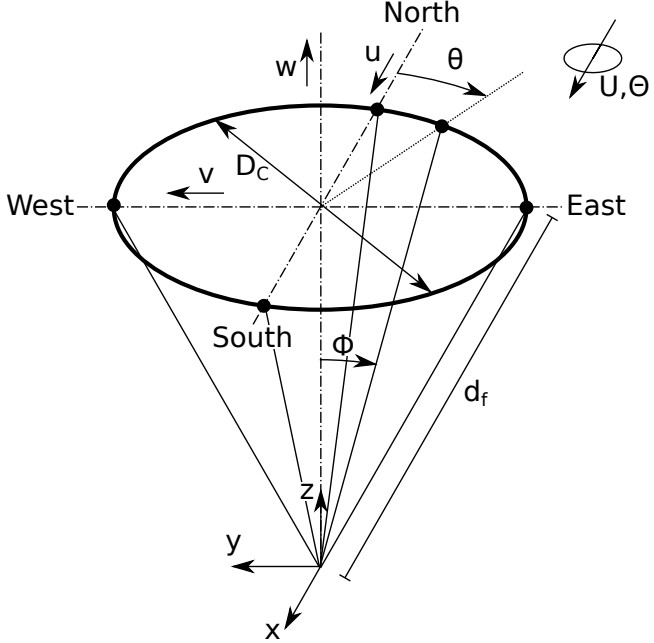

**Figure 1.** Lidar geometry definitions and coordinate system

This research has several practical applications. The reliable elimination of cross-contamination and averaging along the measurement circle would lead to a reduction of the systematic error of wind lidar measurements that is dependent on the prevailing wind conditions and the measurement 60 height. In particular, estimations of the time scale of turbulence could be done with higher certainty, which would support future boundary layer research by means of profiling wind lidars. In addition, estimating the energy content of the wind components at specific wave numbers with higher cer-65 tainty could also help to better predict the operational wind loads of wind turbines and other structures.

Section 2 summarizes the VAD scanning process and describes in detail the averaging and cross-contamination effects it implies for the measurement of turbulence. In Sect. 3 70 the suggested modified data processing methods are described before they are modelled alongside the conventional processing in Sect. 4. The measurements are described in Sect. 5 before the results are compared with the model predictions in Sect. 6. Sect. 7 concludes with the most important 75 findings.

## 2 Lidar theory

### 2.1 Coordinate system and preliminaries

Figure 1 shows the measurement circle of diameter $D_C$ of a VAD scanning lidar and how it is created by the laser beams 80 that are deflected from the zenith by the half cone opening angle $\phi$ and rotate around the zenith with continuously

changing azimuth angle $\theta$. The beams are focused at a point at distance $d_f$ from the lidar which is located at the origin of a three-dimensional left-handed coordinate system. Five of the laser beams are depicted, four in the cardinal directions and one with an arbitrary azimuth angle. The mean wind direction $\Theta$ determined from 10-minute intervals is zero when the wind blows from north to south. The wind vector

$$\boldsymbol{u} = \begin{pmatrix} u \\ v \\ w \end{pmatrix} \qquad (1)$$

is composed of the wind components $u$, $v$ and $w$ that are aligned with the axes of the coordinate system when $\Theta = 0°$. Reynolds decomposition is used for the description of the wind field so that

$$\boldsymbol{u} = \boldsymbol{U} + \boldsymbol{u'} \qquad (2)$$

where $\boldsymbol{u'}$ are the wind speed fluctuations in all three directions and $\boldsymbol{U}$ is the mean wind velocity vector.

## 2.2 Taylor's frozen turbulence hypothesis

The frozen turbulence hypothesis published by Taylor (1938) assumes that turbulence is advected by the mean wind velocity $U$ into the mean wind direction $\Theta$. During the transport process the turbulence remains unchanged, i.e., turbulence measured at one point in space gives information about the turbulence found further downwind some time later. That means for a velocity vector field $\boldsymbol{u}$ when $\boldsymbol{U}$ is aligned with the $x$-axis that

$$\boldsymbol{u}(x,y,z,t) = \boldsymbol{u}(x - Ut, y, z, 0) \qquad (3)$$

The hypothesis is widely used and it is known from experiments that the assumption of frozen turbulence is valid to a high degree for large eddies. For example, Schlipf et al. (2010) measured the inflow velocities of an operating wind turbine at different distances from the rotor plane in order to test the hypothesis of frozen turbulence. They found it to be valid for large scale wind fluctuations with wave numbers $k > 1.25 \times 10^{-1}\,\mathrm{m}^{-1}$. Willis and Deardorff (1976) show that the hypothesis lacks validity when

$$\sigma_u / U > 0.5. \qquad (4)$$

This implies that the validity of the hypothesis depends on the amount of turbulence and that a high degree of validity is expected when the velocity variance is low compared to the mean wind speed.

## 2.3 VAD measurement principle

Continuous-wave wind lidars continuously emit a focused infrared laser beam into the air and detect the small portion of the radiation that is backscattered by particles along the beam path towards the beam's origin. The velocity of the backscattering particles relative to the beam direction is then determined by analyzing the Doppler shift between the frequencies of outgoing and incoming radiation. It is assumed that the backscatterers are lightweight enough to move with the instantaneous wind speed $\boldsymbol{u}$. The measured radial line-of-sight velocities $v_r$ are hence equal to the wind velocity projected onto the beam direction. In order to estimate the three-dimensional wind vector $\boldsymbol{u}$, a minimum of three independent line-of-sight measurements from different directions must be combined.

When VAD scanning is used, the beam is deflected by a wedge prism by a constant half cone opening angle $\phi$ from the zenith and rotated around the zenith with steadily changing azimuth angle $\theta$. Many radial velocities $v_r$ are acquired during one full rotation of the prism. For example in the case of the ZX 300 (previously ZephIR 300), $N = 49$ Doppler spectra are calculated and used to determine the same number of radial velocities. All of them are used to reconstruct one wind vector by applying a least squares fit to

$$v_r = |A\cos(\theta - B) + C| \qquad (5)$$

where the best fit parameters $A$, $B$ and $C$ represent the wind data according to

$$\begin{aligned} v_{hor} &= A/\sin(\phi) \\ \Theta &= B \pm 180° \\ v_{ver} &= C/\cos(\phi) \end{aligned} \qquad (6)$$

The sign of the radial velocity is usually unknown. We are thus faced with a directional ambiguity of $\pm 180°$, but this does not affect the turbulence analysis here. The wind data $v_{hor}$, $\Theta$ and $v_{ver}$ can be translated into wind vectors $\boldsymbol{u}$ easily.

The wind velocity estimations that result from this processing underlie several effects that distinguish them from one-point measurements. These effects can be divided into:

- Averaging
  - along the lines-of-sight
  - along the measurement circle and

- Cross-contamination
  - due to longitudinal separation
  - due to lateral separation

## 2.4 Averaging effects

### 2.4.1 Line-of-sight averaging

In situ wind speed measurements taken with cup anemometers or ultrasonic anemometers have a small measurement volume that can be considered a point. Lidar measurements, in contrast, sense wind velocities along an extended stretch of

the line-of-sight of the laser beam. In the case of continuous-wave lidars, the laser beam leaves the lidar optics with a diameter that corresponds to its effective aperture size $a_0$ and is focused onto a focus point. The distance between the lidar optics and the focus point is the focal distance $d_f$. The signal of the backscattered radiation though originates from anywhere along the illuminated beam, according to a distribution function that has its maximum at the focus point and is proportional to the intensity of the laser light along the beam (Sonnenschein and Horrigan, 1971).

A definite range gate, such as for pulsed lidars, is therefore not applicable to continuous-wave lidars. Instead, the Rayleigh length $l_R$ is a measure of the distance between the focus point and the point at which the cross section of the beam has twice the area of the cross section at the focus point. According to Harris et al. (2006), it is given by

$$l_R = \frac{\lambda d_f{}^2}{\pi a_0{}^2} \tag{7}$$

where $\lambda$ is the laser wavelength and $a_0$ is the effective aperture diameter. The Rayleigh length is quadratically proportional to the focal distance $d_f$ that increases linearly with the selected measurement height level. The degree of line-of-sight averaging is thus strongly dependent on the measurement height level and is higher for larger heights. The values of $l_R$, $a_0$ and $d_f$ for the lidar used in our experiments are given in Table 1.

The intensity of backscattered radiation is a function of the distance $s$ from the focus point along the beam. It is sufficiently well approximated by a Lorentzian function,

$$F(s) = \frac{l_R/\pi}{s^2 + l_R{}^2} \tag{8}$$

where $s$ is the distance from the focus position (Mikkelsen, 2009).

All Doppler spectra that are retrieved during the radial velocity acquisition time are averaged, and the focus point sweeps over a considerable arc of the measurement circle during this time. This arc length $l_A$ is

$$l_A = \frac{D_C \pi}{N} \tag{9}$$

where $N$ is the number of line-of-sight measurements $v_r$ taken during one rotation. In experimental data, the arc averaging effect is contained in the radial velocities. In the models here, we account for this by averaging along the measurement circle.

The Doppler spectra of each line-of-sight measurement resemble the probability density function of the radial wind velocities along the line-of-sight (Branlard et al., 2013). But by determining one single velocity value for each line-of-sight measurement, the turbulence information they contain is filtered out.

The additional temporal averaging along the lines-of-sight is very low, as one measurement takes only $\frac{1}{N}$s. The effect of line-of-sight averaging is very strong for high wave numbers but has some effect on long turbulent structures as well. The effect of line-of-sight averaging is considered in the numerical models and the discussion in this study. But none of the presented data processing methods can avoid the line-of-sight averaging effect.

### 2.4.2 Measurement circle averaging

As described in Sect. 2.3 lidars use all measurement data of at least one full rotation of the prism to reconstruct one wind vector. The resulting system of equations is overdetermined, and in order to find a solution a quadratic best fit is applied. The more lines-of-sight velocities that are used to reconstruct a wind vector, the stronger the averaging and thereby the larger the loss of turbulent kinetic energy in the measurement data. The residual of the best fit is a measure of the degree of this form of averaging but is usually not used in the processing.

The diameter $D_C$ of the measurement circle is

$$D_C = 2h \tan\phi \tag{10}$$

with $h$ being the measurement height and $\phi$ the half cone opening angle. The spatial separation between the points that one reconstructed wind vector is composed of, thus linearly increases with measurement height. The larger the cone diameter, the stronger the circle averaging. Turbulence with a length scale below the diameter of the averaging circle is affected the most.

In addition to the spatial separation of the measurement points along the measurement circle, the acquisition time must be considered. The mean wind motion carries the air while it is probed, which might further increase the separation of measurement points in the mean wind direction. The ZephIR 300 measures one full rotation in one second, and the distance the air moves within this time is usually small compared to $D_C$. The effect of temporal averaging is therefore often small compared to the spatial averaging. One example for the path of measurements that is averaged over is given in Fig. 4a. The circle diameter represents the spatial averaging and the shift along-wind with the speed $U$ represents the temporal averaging.

### 2.5 Cross-contamination

### 2.5.1 Cross-contamination due to longitudinal separation

Another cause for differences in the shape of turbulence spectra from one point measurements and their counterparts from VAD scanning lidars is cross-contamination of different velocity components. VAD scanning lidars combine measurements from spatially separated locations where differing

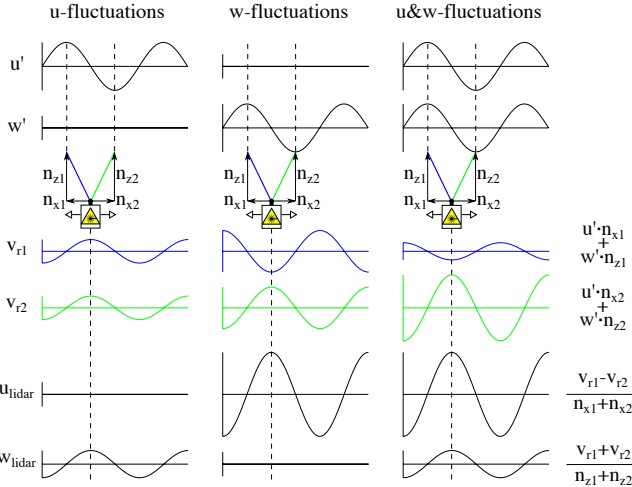

**Figure 2.** Visualization of cross-contamination caused by longitudinal spacing of measurement points 1 and 2. The wavelength of $u'$ and $w'$ equals twice the separation distance of the focus points of the lidar (indicated by box with yellow symbol). The resulting measurement values of the $u$-component are contaminated by fluctuations in $w$-direction and vice versa.

velocities may prevail as if they were collected at one point. This leads to a redistribution of turbulent energy among the velocity components $u$, $v$ and $w$. Lidar-derived spectra of one of the components can at certain wave numbers show lower energy values than the original wind spectrum of that component but may also show too high values due to a contribution from a different velocity component. To better understand cross-contamination we divide the effect into two different types of separations. First we look into longitudinal separations, i.e., separation along the mean wind direction. Fluctuations at two points separated in line with the wind are highly correlated. If the assumption of frozen turbulence is correct, the coherence would be 1 for all separation lengths and all wave numbers. One example of cross-contamination of correlated fluctuations between two longitudinally separated points is visualized in Fig. 2. The chosen wavelength of the wind fluctuations equals twice the separation distance. This can be called the first resonance wavelength. The resonance wavelengths are given by

$$\lambda_{res,n} = \frac{2D_C}{2n-1}. \tag{11}$$

The corresponding resonance wave numbers are

$$k_{res,n} = \frac{(2n-1)\pi}{D_C} \tag{12}$$

where n=1,2,3... The resulting values for the first two resonance points are given in Table 1.

The two beam directions in line with and against the mean wind direction can be used to determine $u_{lidar}$ and $w_{lidar}$ by using the formulas on the right hand side of the figure. This example looks at these two lines-of-sight. The $v$-component can be ignored because transverse fluctuations are not detected by the upstream and downstream beams. The example demonstrates a case with isotropic turbulence, i.e., arbitrary but identical amplitudes for fluctuations in all orientations. Averaging along the lines-of-sight is ignored here for simplicity. The first column of graphs in the figure isolates the $u$-fluctuations $u'$ and shows the resulting lidar measured signal for the two radial velocities in upwind and downwind direction, i.e., $v_{r1}$ and $v_{r2}$. When these two signals are combined in the usual way, the reconstructed wind speed components $u'_{lidar}$ and $w'_{lidar}$ differ strongly from the real inflow conditions $u'$ and $w'$. The lidar is blind to wind speed fluctuations in $u$-direction and instead attributes the fluctuations to some extent to the estimation of $w'_{lidar}$. The same is done for $w'$ in the second column, and the resulting effect is the reverse. The vertical fluctuations $w'$ are interpreted solely as amplified fluctuations of $u'_{lidar}$.

The last column combines the two previous cases and shows the resulting distribution of amplitudes which depends on the half cone opening angle $\phi$. When $\phi < 45°$ the lidar is more sensitive to vertical variations than to horizontal ones, and the contamination of $u'$ caused by $w'$ is more severe than vice versa.

In a more realistic situation, turbulence is non-isotropic and the amplitude of $w'$ at this first resonance wave number is often considerably lower than the amplitude of $u'$ which leads to a different distribution of contamination which can be estimated as follows. We use equations 31 and 33 to define the lidar-derived variance in $u$-direction

$$\sigma_{u,lidar}^2 = \left\langle \left( \frac{\Delta v}{-2\sin\phi} \right)^2 \right\rangle. \tag{13}$$

In general, the differences of the line-of-sight velocities aligned with the mean wind $\Delta v$ contain contributions from wind fluctuations in $u$ and $w$-direction $\Delta v_u$ and $\Delta v_w$ respectively. Here we look at the resonance case where $\Delta v_u = 0$ and thus $\Delta v = \Delta v_w$. We get

$$\sigma_{u,lidar,res}^2 = \left\langle \left( \frac{\Delta v_w}{-2\sin\phi} \right)^2 \right\rangle = \left\langle \left( \frac{2w'\cos\phi}{-2\sin\phi} \right)^2 \right\rangle \tag{14}$$

$$= \cot^2\phi\, \sigma_{w,res}^2 \approx 2.86\, \sigma_{w,res}^2$$

when $\phi = 30.6°$ as for the lidar we used in this study. The subscript $res$ indicates that the equation is only valid for inflow fluctuations at resonance, as in the example given before.

In Sect. 4 we develop a model to predict lidar-derived spectra. This model was used to create the plots shown in Fig. 3. Fig. 3a shows the modelled spectra of the wind components, $u_{wind}$ and $w_{wind}$, as solid black and red lines. The parameters of the underlying spectral tensor are given in Table 1. They were chosen to best represent the wind conditions

found during the experiment presented in Sect. 6. The model was used to estimate the $u$-component of the wind from two lidar beams that point in the upwind and downwind direc-tions. Also here, we did not include line-of-sight averaging to isolate the effect of cross-contamination. The principle of the setup is the same as explained for Fig. 2 but now we see results for all inflow wave numbers and use anisotropic turbu-lence. The resulting lidar-derived spectrum $u_{lidar,sum}$ of the $u$-component of the wind is the sum of the lidar's interpre-tation of the wind components $u_{lidar,u}$ and $u_{lidar,w}$. We see that the lidar estimated spectrum of $u_{lidar,sum}$ lies a bit be-low the target spectrum of $u_{wind}$ for most wave numbers but not at the first and second resonance points that are marked with grey dashed vertical lines. There it exceeds the target spectrum. The reason becomes apparent when we look at the components $u_{lidar,u}$ and $u_{lidar,w}$ that $u_{lidar,sum}$ is com-posed of. We find that the lidar sees $u_{wind}$ nearly to its full extend for very low wave numbers but when we come close to the resonance points $u_{lidar,u}$ drops to zero. The contribu-tion of the vertical wind $u_{lidar,w}$ shows a mirrored behavior and is amplified according to Eq. 14 since $\phi < 45°$.

### 2.5.2   Cross-contamination due to lateral separation

When the lines-of-sight under consideration are not longitu-dinally but laterally separated, they do not face resonance but instead a second form of cross-contamination. The strength of the contamination depends then on the coherence of the turbulence for the given lateral separation. When the fluc-tuations at the two selected focus points are very coherent i.e., their correlation is close to unity, we can expect that the lidar-derived wind speed estimates are correct and no cross-contamination occurs. This can be observed at very low wave numbers where a high degree of coherence is expected. The other extreme is found at the other end of the spectrum where small fluctuations measured at both focus points are uncor-related. The lidar-derived spectrum is there a linear combi-nation of the variances of the involved components $v$ and $w$ according to

$$\sigma_{v,lidar}^2 = \left\langle \left( \frac{\Delta v}{-2\sin\phi} \right)^2 \right\rangle = \left\langle \left( \frac{\Delta v_v}{-2\sin\phi} \right)^2 \right\rangle + \left\langle \left( \frac{\Delta v_w}{-2\sin\phi} \right)^2 \right\rangle .$$

(15)

In the case of fully uncorrelated fluctuations we know that $\Delta v_v = -v'\sin\phi$ and $\Delta v_w = w'\cos\phi$ and the variance $\sigma_{v,lidar,unc}^2$ of the lidar-derived $v$-velocity is

$$\sigma_{v,lidar,unc}^2 = \left\langle \left( \frac{-v'\sin\phi}{-2\sin\phi} \right)^2 \right\rangle + \left\langle \left( \frac{w'\cos\phi}{-2\sin\phi} \right)^2 \right\rangle$$

(16)

$$= \frac{1}{2} \left( \sigma_{v,unc}^2 + \sigma_{w,unc}^2 \cot^2\phi \right) \approx 0.5\sigma_{v,unc}^2 + 1.43\sigma_{w,unc}^2$$

for the lidar with a half cone opening angle of $\phi = 30.6°$. These two situations and all cases in between are shown in

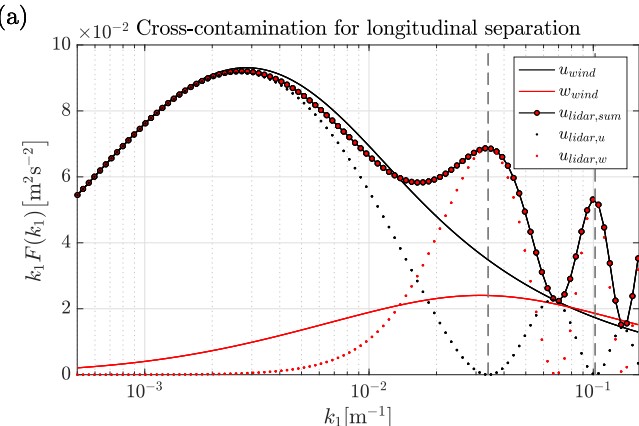

(a)

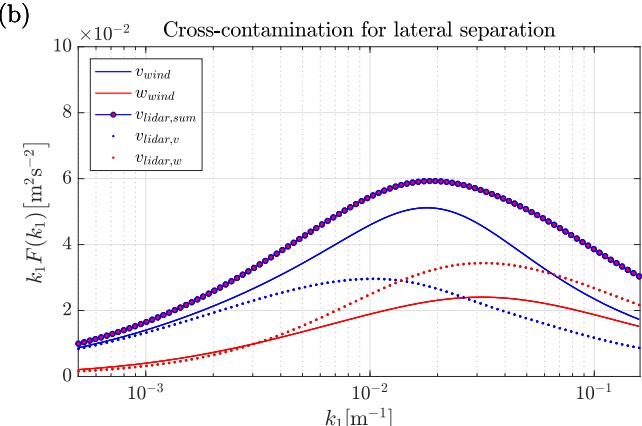

(b)

**Figure 3.** Modelled cross-contamination effect inherent in (a) the $u$-spectrum from two longitudinally separated points with $\Delta x = D_C$ and (b) the $v$-spectrum from two laterally separated points with $\Delta y = D_C$. The solid lines are the spectra of the involved wind com-ponents. The dotted lines show the contribution of these wind com-ponents to the lidar spectra (circle markers). Averaging along the lines-of-sight is excluded.

Fig. 3b. The difference to the plots in Fig. 3a is that the two beams that point into and against the $v$-direction are used here to estimate the $v$-spectrum $v_{lidar,sum}$. The target spec-trum of the $v$-component of the wind $v_{wind}$ is given as well as the $w$-spectrum $w_{wind}$ that contaminates the signal. From the $v_{lidar,v}$ and $v_{lidar,w}$ curves it can be seen that at very low wave numbers hardly any contamination occurs but mainly because the $w$-component $w_{wind}$ itself contains a low energy density at low wave numbers. As it increases for higher wave numbers, the contamination also gets more severe. In this ex-ample $w_{wind}$ dominates the lidar spectrum $v_{lidar,sum}$ for all wave numbers above approximately $k_1 = 1.4 \times 10^{-2}\,\mathrm{m}^{-1}$. The result is that the lidar overestimates the $v$-variances for all wave numbers. Such an effect is also reported by Wyn-gaard (1968). Thus, it is essential for accurate turbulence measurements to minimize spatial separation.

VAD scanning along the whole measurement circle is more complex than using only two beams. Examining the

two beams aligned with or perpendicular to the mean wind direction is not sufficient to fully understand the effect of cross-contamination. For circle scans, all three wind speed components are involved in contaminating all the beams that do not point in the four cardinal directions. We refer to the model presented in Sect. 4.1 and especially Eqs. (24), (25) and (26) of the spectral weighting functions therein to better understand which components influence another.

The lidar can also be configured to perform a so-called three-second scan, in which one measurement cycle is built from data from three full rotations. This limits the cross-contamination but comes at the cost of much stronger averaging along the measurement circle, especially in strong wind cases, and three times slower sampling rate. The ability to measure turbulence with this approach is so weak that it is not further investigated in this paper. Instead, the next chapter suggests two methods that can be used to reduce both averaging and cross-contamination.

## 3 Modified data processing

### 3.1 Squeezed measurement circles

In conventional VAD data processing, each measurement cycle consists of the radial velocities that are acquired during one full rotation of the prism. The data used in the reconstruction of one wind vector originates thus from an air volume with the shape of a cone with a diameter of $D_C$ at the height of focus. This results in the above mentioned cross-contamination effects.

One way to eliminate the cross-contamination due to longitudinal separation and mitigate the averaging along the measurement circle lies in making use of Taylor's frozen turbulence hypothesis. As mentioned in Sect. 2.2, the hypothesis assumes that turbulent structures are transported by the mean wind motion without changing. This implies that all turbulent structures that enter the measurement cone at one time are identical after some time $t$ when they leave the cone. The time it takes to cross the measurement circle can be estimated for all azimuth directions $\theta$ by

$$t(\theta) = \cos\theta \frac{D_C}{U} \qquad (17)$$

where $U$ is the mean wind velocity calculated from conventional VAD processing.

The basic idea here is to introduce a time lag $\tau = t$ into the data processing so that each air package that is involved in the reconstruction of one wind vector is scanned twice: once when it enters and again when it leaves the measurement cone. The composition of the measurement circles is shown in Fig. 4 from a coordinate system that is moving with the mean wind $U$. In this example $D_C = 92.3\,\mathrm{m}$ and $U = 19.5\,\mathrm{m\,s}^{-1}$. With conventional VAD data processing, the measurement circle is made up of all $N$ consecutive measurements from one cycle (red segment). By contrast, the

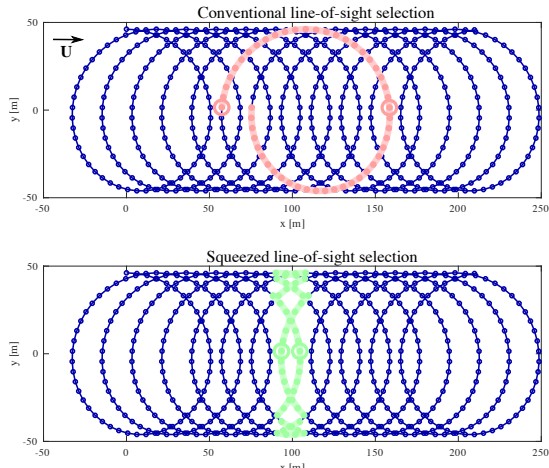

**Figure 4.** Selection of line-of-sight measurements for the reconstruction of one wind vector for when (top, in red) conventional VAD processing and (bottom, in green) the method of squeezed measurement circles is applied. Within the red and green segments, small red and green rings indicate the particular beams selected for 2-beam processing. In this example, $U = 19.5\,\mathrm{m\,s}^{-1}$, $D_C = 92.3\,\mathrm{m}$ and $f_S = 1\,\mathrm{Hz}$.

lower part of Fig. 4 illustrates the introduction of the time delay $\tau$, in which line-of-sight measurements from a total of $M = 6$ different measurement cycles are combined to estimate one wind vector (green segments). In other words, with conventional data processing, a measurement cycle is composed of volumes which are widely spatially distributed. The proposed new method picks measurement data taken from what we term a squeezed measurement circle (SMC).

A restriction that comes with the idea of squeezing is that the circle sample rate $f_S$ must be high enough to be able to select measurements that were acquired with a time difference reasonably close to $\tau$. That drastically limits the amount of measurement heights that should be selected, especially in strong wind cases. For the measurements analyzed in this paper, the lidar scanned continuously at only one height level, which in general makes sense to measure turbulence effectively.

### 3.2 2-beam method

The conventional method of averaging data from all available lines-of-sight to reconstruct 3-dimensional wind vectors leads to strong averaging along the measurement circle. The method is known to deliver reliable values for the mean wind speed and direction. The directional information allows it to determine the two beams that lie in the upstream and downstream directions. Within the red and green segments of Fig. 4, small red and green rings indicate these particular beams. These two beams can in a second processing step be used to

estimate the $u$- and $w$-components of the wind vectors for turbulence estimations. The resulting values are then not averaged along the measurement circle. This is comparable to the DBS method in cases where the mean wind blows in line with two of the lines-of-sight. But an advantage of the 2-beam method over the DBS strategy is that the relative angle between the mean wind and the two beams is kept constant in any prevailing wind direction. This is an advantage since beams pointing upwind and downwind are immune to contamination by the cross wind component $v$.

When the 2-beam method is combined with the idea of squeezing, then measurements of the $u$- and $w$-components are taken at virtually one focus point following the flow. Only the line-of-sight averaging and some minor longitudinal separation between the different locations along the two beams remain.

That is unfortunately not true when estimating the $v$-component of turbulence. Instead, several problems occur. Intuitively, one would choose a beam direction perpendicular to the mean wind direction in order to estimate the $v$-component of the wind. But the radial velocities in this line-of-sight direction are often close to zero, and such estimates from continuous-wave lidars are usually not reliable (Mann et al., 2010; Dellwik et al., 2010). The transverse $v$-component must therefore be estimated either by VAD/SMC processing or by selecting a different third beam direction. In the latter case the results would then be influenced by contamination not only from $w$ but also from the $u$-component. This lies outside the scope of this study. Therefore no $v$-data from measurements are processed with the 2-beam method.

Like conventional VAD processing, the SMC method and 2-beam method require a wind field that is statistically homogeneous in the horizontal directions to yield correct results.

## 4   Description of the model

The mathematics of deducing the lidar-measured spectrum from the second-order statistics of turbulence is very convoluted. Therefore, we make the assumption that the measurements are done much faster than it takes the air to move from one side of the scanning circle to the other, i.e., we assume that $\frac{1}{f_S} \ll \tau$. Effectively, the scanning circle is measured continuously. It is difficult to assess the magnitude of the error committed by the assumption of continuous measurements, but we assume it is negligible.

### 4.1   VAD and SMC

In order to model spectra obtained from conventionally VAD processed lidar data we closely follow Sathe et al. (2011). They use the geometry of the lidar scan and its along beam weighting function together with information on the spatial structure of surface-layer turbulence (Mann, 1994). The focus point of the lidar is at a distance $d_f$ away in the direction

given by the unit vector

$$\boldsymbol{n}(\theta) = (-\cos\theta\sin\phi, -\sin\theta\sin\phi, \cos\phi) \tag{18}$$

where $\theta$ is the azimuth angle and $\phi$ is the half cone opening angle. The line-of-sight or radial wind speed that the lidar is measuring is modelled as

$$v_r(\theta, x) = \int_{-\infty}^{\infty} \varphi(s)\boldsymbol{n}(\theta) \cdot \boldsymbol{u}((s+d_f)\boldsymbol{n}(\theta) + x\boldsymbol{e}_1)ds \ . \tag{19}$$

where $\varphi$ is the spatial weighing function of the continuous-wave lidar that we assume to be a Lorentzian function with the Rayleigh length $l_R$. $\boldsymbol{u}$ is the three-dimensional velocity field suppressing the time argument since we are assuming Taylor's hypothesis. The integration variable $s$ is the distance along the beam from the focus point. The dot product assures that we get the line-of-sight velocity. We use $x$, the coordinate aligned with the mean wind vector, instead of time. $\boldsymbol{e}_1$ is the unit vector aligned with $x$.

The $w$-, $u$-, and $v$-components of the velocity are calculated by the first three Fourier coefficients of $v_r$ as a function of $\theta$, i.e $w$ is calculated from

$$A(x) = \frac{1}{2\pi} \int_0^{2\pi} v_r(\theta, x)d\theta \tag{20}$$

In Sathe et al. (2011) variances are calculated for a conically scanning continuous-wave lidar and it is trivial to extend that to spectra. Spectra were in fact calculated in Sathe and Mann (2012) but only for a pulsed system. In Sathe et al. (2011) the variances for a conically scanning continuous-wave system, e.g., a ZephIR 300 (Smith et al., 2006; Kindler et al., 2007), were given by

$$\langle w^2\rangle\cos^2\phi = \int \Phi_{ij}(\boldsymbol{k})\alpha_i(\boldsymbol{k})\alpha_j^*(\boldsymbol{k})d\boldsymbol{k} \tag{21}$$

$$\langle u^2\rangle\sin^2\phi = \int \Phi_{ij}(\boldsymbol{k})\beta_i(\boldsymbol{k})\beta_j^*(\boldsymbol{k})d\boldsymbol{k} \tag{22}$$

$$\langle v^2\rangle\sin^2\phi = \int \Phi_{ij}(\boldsymbol{k})\gamma_i(\boldsymbol{k})\gamma_j^*(\boldsymbol{k})d\boldsymbol{k} \tag{23}$$

where $^*$ means complex conjugation. The spectral weighting functions $\alpha$, $\beta$ and $\gamma$ are

$$\alpha_i(\boldsymbol{k}) = \frac{1}{2\pi} \int_0^{2\pi} n_i(\theta)\mathrm{e}^{\mathrm{i}d_f\boldsymbol{k}\cdot\boldsymbol{n}(\theta)}\mathrm{e}^{-l|\boldsymbol{k}\cdot\boldsymbol{n}(\theta)|}d\theta \tag{24}$$

$$\beta_i(\boldsymbol{k}) = \frac{1}{\pi} \int_0^{2\pi} \cos\theta\, n_i(\theta)\mathrm{e}^{\mathrm{i}d_f\boldsymbol{k}\cdot\boldsymbol{n}(\theta)}\mathrm{e}^{-l|\boldsymbol{k}\cdot\boldsymbol{n}(\theta)|}d\theta \tag{25}$$

$$\gamma_i(\boldsymbol{k}) = \frac{1}{\pi} \int_0^{2\pi} \sin\theta\, n_i(\theta)\mathrm{e}^{\mathrm{i}d_f\boldsymbol{k}\cdot\boldsymbol{n}(\theta)}\mathrm{e}^{-l|\boldsymbol{k}\cdot\boldsymbol{n}(\theta)|}d\theta \ . \tag{26}$$

The spectra measured by the conically scanning lidar will be

$$\cos^2\phi\, F_w^Z(k_1) = \hat{T}_f(k_1) \iint\limits_{-\infty}^{\infty} \Phi_{ij}(\boldsymbol{k})\alpha_i(\boldsymbol{k})\alpha_j^*(\boldsymbol{k}) dk_2 dk_3 \tag{27}$$

and likewise for the $u$- and $v$-components. The function

$$\hat{T}_f(k_1) = \text{sinc}^2\left(\frac{k_1 L_f}{2}\right) \tag{28}$$

where $\text{sinc}\, x = \frac{\sin x}{x}$ is multiplied to the integral to account for the finite time of circle scanning before a velocity estimate is obtained. $L_f$ is the mean wind speed multiplied with this finite time (see Sathe et al. (2011) for details).

To apply the method of squeezing and model the spectra we get from SMC processing, we now substitute Eq. (19) with

$$\tilde{v}_r(\theta, x) = \int\limits_{-\infty}^{\infty} \varphi(s)\boldsymbol{n}(\theta)\cdot\boldsymbol{u}((s+d_f)\boldsymbol{n}(\theta)+(x-d_f n_1(\theta))\boldsymbol{e}_1) ds\ . \tag{29}$$

Following the exact same steps as in Sathe et al. (2011) but using Eq. (29) instead of (19) we arrive at Eqs. (21) – (23) but with the complex exponential in (24) – (26) exchanged with

$$e^{\mathrm{i}d_f(\boldsymbol{k}\cdot\boldsymbol{n}(\theta)-k_1 n_1(\theta))} \tag{30}$$

### 4.2 2-beam method

Only the up- and downwind beams to determine the $u$ and $w$ components of the wind vector could introduce less averaging than using the whole circle.

When the mean wind is blowing from the north, the unit vectors in the up- and downwind directions are called $\boldsymbol{n}^u$ and $\boldsymbol{n}^d$, respectively. Their unit vectors are

$$\boldsymbol{n}^u = (-\sin\phi, 0, \cos\phi) \tag{31}$$

and with the opposite sign on the first component for $\boldsymbol{n}^d$.

Parallel to Eq. (19) the line-of-sight velocity measured by the upwind beam is assumed to be

$$v^u(x) = \int\limits_{-\infty}^{\infty} \varphi(s)\boldsymbol{n}^u\cdot\boldsymbol{u}(s\boldsymbol{n}^u+d_f\boldsymbol{n}^u+x\boldsymbol{e}_1) ds \tag{32}$$

The $u$-component estimated by the lidar is normally

$$u_{\text{lidar}} = \frac{\Delta v}{n_1^u - n_1^d} \tag{33}$$

where

$$\Delta v = v^u(x) - v^d(x)$$
$$= \int\limits_{-\infty}^{\infty} \varphi(s)\big[\boldsymbol{n}^u\cdot\boldsymbol{u}((s+d_f)\boldsymbol{n}^u+x\boldsymbol{e}_1) \tag{34}$$
$$\qquad - \boldsymbol{n}^d\cdot\boldsymbol{u}((s+d_f)\boldsymbol{n}^d+x)\big] ds \tag{34}$$

The correlation function of $\Delta v$ is

$$R_{\Delta v}(r) = \langle\Delta v(x)\Delta v(x+r)\rangle$$
$$= \iint\limits_{-\infty}^{\infty} \varphi(s)\varphi(s')$$
$$\times\Big\langle \big[\boldsymbol{n}^u\cdot\boldsymbol{u}((s+d_f)\boldsymbol{n}^u+x\boldsymbol{e}_1)$$
$$\qquad - \boldsymbol{n}^d\cdot\boldsymbol{u}((s+d_f)\boldsymbol{n}^d+x\boldsymbol{e}_1)\big]$$
$$\times \big[\boldsymbol{n}^u\cdot\boldsymbol{u}((s'+d_f)\boldsymbol{n}^u+(x+r)\boldsymbol{e}_1)$$
$$\qquad - \boldsymbol{n}^d\cdot\boldsymbol{u}((s'+d_f)\boldsymbol{n}^d+(x+r)\boldsymbol{e}_1)\big]\Big\rangle dsds'\ . \tag{35}$$

Expanding the product inside the ensemble average ($\langle\rangle$) and using the definition of the correlation tensor of the velocity field, $R_{ij}(\boldsymbol{r}) \equiv \langle u_i(\boldsymbol{x})u_j(\boldsymbol{x}+\boldsymbol{r})\rangle$ one gets

$$R_{\Delta v}(r) = \iint\limits_{-\infty}^{\infty} \varphi(s)\varphi(s')\times \tag{36}$$
$$\Big\{ n_i^u n_j^u R_{ij}((-s+s')\boldsymbol{n}^u+r\boldsymbol{e}_1)$$
$$+ n_i^d n_j^d R_{ij}((-s+s')\boldsymbol{n}^d+r\boldsymbol{e}_1)$$
$$- n_i^u n_j^d R_{ij}(s'\boldsymbol{n}^d-s\boldsymbol{n}^u+d_f(\boldsymbol{n}^d-\boldsymbol{n}^u)+r\boldsymbol{e}_1)$$
$$- n_i^d n_j^u R_{ij}(s'\boldsymbol{n}^u-s\boldsymbol{n}^d+d_f(\boldsymbol{n}^u-\boldsymbol{n}^d)+r\boldsymbol{e}_1)\Big\} dsds'$$

Now we use the relation between the velocity covariance tensor and the spectral velocity tensor

$$R_{ij}(\boldsymbol{r}) = \int \Phi_{ij}(\boldsymbol{k})\exp(\mathrm{i}\boldsymbol{k}\cdot\boldsymbol{r}) d\boldsymbol{k} \tag{37}$$

where $\int d\boldsymbol{k} \equiv \iiint_{-\infty}^{\infty} dk_1 dk_2 dk_3$, to express the auto-covariance function as

$$R_{\Delta v}(r) = \iint\limits_{-\infty}^{\infty} \varphi(s)\varphi(s')\times \tag{38}$$
$$\Big\{ n_i^u n_j^u \int \Phi_{ij}(\boldsymbol{k})\exp\big(\mathrm{i}\boldsymbol{k}\cdot((-s+s')\boldsymbol{n}^u+r\boldsymbol{e}_1)\big) d\boldsymbol{k}$$
$$+ n_i^d n_j^d \int \Phi_{ij}(\boldsymbol{k})\exp\big(\mathrm{i}\boldsymbol{k}\cdot((-s+s')\boldsymbol{n}^d+r\boldsymbol{e}_1)\big) d\boldsymbol{k}$$
$$- n_i^u n_j^d \int \Phi_{ij}(\boldsymbol{k})\exp\big(\mathrm{i}\boldsymbol{k}\cdot(s'\boldsymbol{n}^d-s\boldsymbol{n}^u+d_f(\boldsymbol{n}^d-\boldsymbol{n}^u)+r\boldsymbol{e}_1)\big) d\boldsymbol{k}$$
$$- n_i^d n_j^u \int \Phi_{ij}(\boldsymbol{k})\exp\big(\mathrm{i}\boldsymbol{k}\cdot(s'\boldsymbol{n}^u-s\boldsymbol{n}^d+d_f(\boldsymbol{n}^u-\boldsymbol{n}^d)+r\boldsymbol{e}_1)\big) d\boldsymbol{k}\Big\} dsds'$$

By interchanging the order of integration of $\boldsymbol{k}$ and the $s$'s we can cast the expression in terms of the Fourier transform of $\varphi$ which in the case of a Lorentzian is $\hat{\varphi}(k) = \exp(-l_R|k|)$. Thereafter, we Fourier transform $R_{\Delta v}$ with respect to $r$ to get the spectrum $F_{\Delta v}(k_1)$. After that process the first term in (38) becomes

$$n_i^u n_j^u \int \Phi_{ij}(\boldsymbol{k}) \hat{\varphi}(\boldsymbol{k} \cdot \boldsymbol{n}^u) \hat{\varphi}(\boldsymbol{k} \cdot \boldsymbol{n}^u) dk_2 dk_3$$

and upon rearrangement we finally obtain

$$
\begin{aligned}
F_{\Delta v,n}(k_1) = \int \Phi_{ij}(\boldsymbol{k}) \bigg\{ & n_i^u n_j^u \left|\hat{\varphi}(\boldsymbol{k} \cdot \boldsymbol{n}^u)\right|^2 + n_i^d n_j^d \left|\hat{\varphi}(\boldsymbol{k} \cdot \boldsymbol{n}^d)\right|^2 \\
& - 2 n_i^u n_j^d \hat{\varphi}(\boldsymbol{k} \cdot \boldsymbol{n}^u) \hat{\varphi}(\boldsymbol{k} \cdot \boldsymbol{n}^d) \qquad (39) \\
& \times \cos\left(d_f \boldsymbol{k} \cdot (\boldsymbol{n}^d - \boldsymbol{n}^u)\right) \bigg\} dk_2 dk_3 \; .
\end{aligned}
$$

The derivation of the "squeezed" spectrum is parallel to the normal spectrum. The only difference lies in the definition of $\Delta v$. Now we define it as

$$\Delta v_s(x) = v^u(x - n_1^u d_f) - v^d(x - n_1^d d_f) \qquad (40)$$

Using the exact same steps that led to (39), we see that the cosine term in that equation has to be substituted with one and we get

$$
\begin{aligned}
F_{\Delta v,s}(k_1) = \int \Phi_{ij}(\boldsymbol{k}) \bigg\{ & n_i^u n_j^u \left|\hat{\varphi}(\boldsymbol{k} \cdot \boldsymbol{n}^u)\right|^2 + n_i^d n_j^d \left|\hat{\varphi}(\boldsymbol{k} \cdot \boldsymbol{n}^d)\right|^2 \\
& - 2 n_i^u n_j^d \hat{\varphi}(\boldsymbol{k} \cdot \boldsymbol{n}^u) \hat{\varphi}(\boldsymbol{k} \cdot \boldsymbol{n}^d) \bigg\} dk_2 dk_3 \; . \\
& \qquad\qquad\qquad\qquad\qquad\qquad\qquad (41)
\end{aligned}
$$

To obtain the spectrum of $u$, $F_{\Delta v,(s)}$ simply has to be divided by $(n_1^u - n_1^d)^2$ according to (33).

When obtaining the spectrum of $v$, we simply exchange the unit vectors of the up- and downwind beams $\boldsymbol{n}^u$ and $\boldsymbol{n}^d$ in all equations by the values of the west- and eastbound beams $\boldsymbol{n}^w$ and $\boldsymbol{n}^e$. In order to obtain the spectrum of $w$, $\Delta v$ defined in Eq. (34) has to be replaced by the sum of both radial velocities $v^u(x) + v^d(x)$ and $F_{\Delta v,(s)}$ must eventually be divided by $(n_3^u + n_3^d)^2$.

To compare the different methods to calculate spectra from a lidar Eqs. (39) and (41) have to be evaluated with a model for the spectral tensor. We chose the spectral tensor from Mann (1994) and select the model parameters so that the model spectra resemble the spectra from available sonic measurements. The selected parameters are: $L = 65\,\mathrm{m}$, $\Gamma = 4$ and $\alpha\epsilon^{\frac{2}{3}} = 0.023\,\mathrm{m}^{\frac{4}{3}}\mathrm{s}^{-2}$. The unfiltered $u$ target model spectrum that we will later compare the model results against is given by

$$F_u(k_1) = \int \Phi_{11}(\boldsymbol{k}) dk_2 dk_3 \qquad (42)$$

and respectively for the second and third wind component. The model was tested by comparing the theoretical spectra with results from processing computer generated wind field turbulence data (Mann, 1998) and was found to predict all four data processing methods i.e., VAD, SMC, 2-beam and squeezed 2-beam accurately for all 3 wind speed components.

## 5 Description of the measurements

### 5.1 Test site and instrumentation

The test data were collected at the Danish National Test Center for Large Wind Turbines at Høvsøre. The test site is located in West Jutland, Denmark, $1.7\,\mathrm{km}$ east of the North Sea. Apart from the dunes along the coastline, the terrain is nearly flat. The Høvsøre meteorological mast is located to the south of a row of five wind turbines. The reference data were acquired with a Metek USA-1 sonic anemometer that is mounted at $80.5\,\mathrm{m}$ height above the ground. It is attached to a $4.3\,\mathrm{m}$ long boom pointing north. Mast effects can be observed when the wind is blowing from the south. Turbine wake effects influence the measurement signal when the wind blows from the north. For the data set in this study, the inflow is undisturbed. A detailed description of the test site is given in Peña et al. (2016).

Collocated with the meteorological mast, the lidar measurements were taken by a Qinetiq lidar that was configured to continuously scan at $78\,\mathrm{m}$ above the ground. The lidar is comparable to the current ZX 300 (previously ZephIR 300) but the effective aperture size is slightly lower which results in a longer Rayleigh length and thus greater line-of-sight averaging. The lidar was equipped with an opto-acoustic modulator that makes it possible to detect the direction of the radial velocities. Line-of-sight velocities calculated from the centroid of the Doppler spectra are used in the data processing. The precision of these lidar measurements is not exactly known but is in general better than 1% (Pedersen et al., 2012).

Measurement data of 32 subsequent 10-minute intervals are used. The data were acquired on 20.11.2008 between 10:30 and 15:50 local time. The mean wind velocity measured by the sonic anemometer during this period varied from $14.2\,\mathrm{m\,s}^{-1}$ to $22.6\,\mathrm{m\,s}^{-1}$ with an average of $19.5\,\mathrm{m\,s}^{-1}$ and a standard deviation of $2.0\,\mathrm{m\,s}^{-1}$. The turbulence intensity varied from 4.7% to 14.0%, with a mean of 8.8% and standard deviation of 2.0%. The wind blew from the northwest and the atmospheric stability was neutral. Table 1 summarizes the most important information about the experimental setup.

### 5.2 Data processing

The time series of all 10-minute intervals derived from all processing methods are used to compute turbulence spec-

| Description | Abbr. | Value | Unit |
|---|---|---|---|
| Measurement height | $h$ | 78 | [m] |
| Half cone angle | $\phi$ | 30.6 | [deg] |
| Cone diameter | $D_C$ | 92.3 | [m] |
| Focus distance | $d_f$ | 90.6 | [m] |
| Prism rotation | $f_S$ | 1 | [Hz] |
| Measurements per cycle | N | 50 | [1] |
| Laser wave length | $\lambda$ | 1550 | [nm] |
| Effec. aperture diam. | $a_0$ | 24 | [mm] |
| Mean wind speed | $U_{mean}$ | 19.5 | [m s$^{-1}$] |
| 1. Resonance | $\lambda_{res1}$ | 184.5 | [m] |
| | $k_{res1}$ | 0.034 | [m$^{-1}$] |
| 2. Resonance | $\lambda_{res2}$ | 61.5 | [m] |
| | $k_{res2}$ | 0.102 | [m$^{-1}$] |
| No. of cycles to cover $\tau$ | M | 0-5 | [1] |
| Rayleigh length | $l_R$ | 7.03 | [m] |
| Full width half maximum | $2l_R$ | 14.07 | [m] |

**Table 1.** Key specifications of the lidar used in the measurements

tra. The measurement rate for the lidar is 1 Hz. Although it would have been possible in the 2-beam processing to calculate measurement values with a rate of 2 Hz by using every newly retrieved radial velocity together with its predecessor, it was decided to use only independent measurements acquired every full second. The sonic anemometer measures with a rate of 20 Hz. This high frequency data is downsampled by the use of the MATLAB function 'resample' to a frequency of 1 Hz. The function includes a low-pass filter to avoid anti-aliasing. The data rate is thus for all methods 1 Hz. The analyzed frequency range from $\frac{1}{600}$ Hz to $\frac{1}{2}$ Hz equals the wave number range from roughly $5.4 \times 10^{-4}$ m$^{-1}$ to $1.6 \times 10^{-1}$ m$^{-1}$. The spectra are then averaged for all intervals and the results are then binned into 30 logarithmically spaced wave number intervals spread across the wave number axis to avoid high density of values and maintain readability towards higher wave numbers.

The effects of de-trending (Hansen and Larsen, 2005) and spike removal (Hojstrup, 1993) on the spectra were both negligible for this dataset, so neither was applied here.

# 6 Discussion of the results

## 6.1 u-spectra

Figure 5 shows the spectra of the $u$-fluctuations for all processing methods from measurement data (triangle markers) and the corresponding model predictions (solid lines). We will first discuss the results from processing the whole measurement circle shown in Fig. 5a, followed by the discussion of the results of the 2-beam method, shown in Fig. 5b.

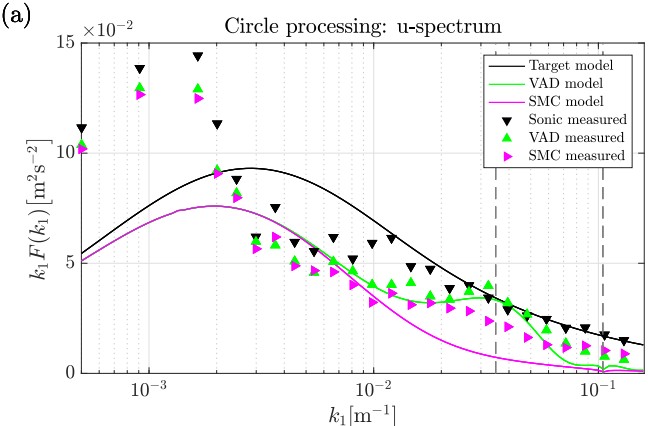

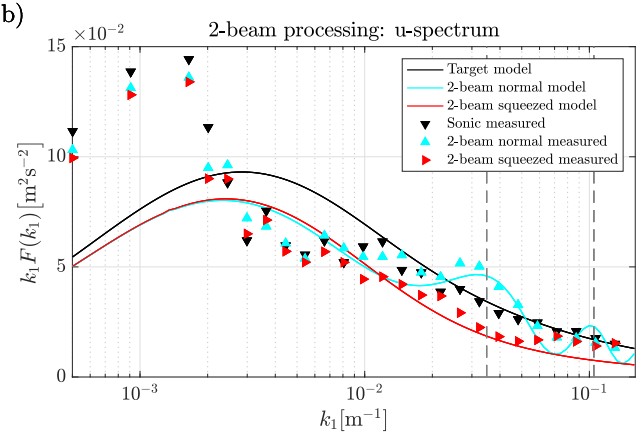

**Figure 5.** Modelled (solid lines) and measured (triangle markers) $u$-spectra from data processing where (a) all radial measurements are used and (b) only two beams are used. Colors correspond to processing method. The grey vertical dashed lines represent the first and second resonance wave number.

### 6.1.1 Circle processing

To begin with, the model predictions of conventional VAD processing and the new SMC method are compared against each other and with regard to the true $u$ target model spectrum acquired from the spectral tensor according to Eq. 42. The model prediction of the conventionally processed VAD lidar data shows some attenuation of the spectral energy even for very low wave numbers. This can be partly explained by the infinitely long tails of the line-of-sight averaging function given in Eq. (8). That means that even very large eddies are slightly weakened by the underlying Lorentzian function. Averaging along the measurement circle might also have some small additional impact on large scale turbulence. Both averaging effects get more and more severe for increasing wave numbers until the measured spectral energy reaches values close to zero at roughly $k_1 = 10^{-1}$ m$^{-1}$ and above. The tendency of increasing attenuation with regard to the target spectrum is interrupted around the first resonance frequency that is indicated by a vertical grey dashed line at

$k_1 = 3.4 \times 10^{-2}\,\mathrm{m}^{-1}$. Here the energy density increases and reaches coincidentally roughly the value of the target spectrum. This behavior is as expected an effect of the cross-contamination with energy from both the $w$-spectrum and to a small extend also from $v$. A resonance effect at the second resonance frequency is hardly pronounced since the energy is nearly fully consumed by the line-of-sight averaging.

The SMC model spectrum predicts a similar shape but without the cross-contamination effect from longitudinal separation. Thus, we find no resonance in the computations. The total variance of the $u$-fluctuations $\sigma^2(u')$ is lower here since less additional energy from the $w$ component is contained in the $u_{SMC}$ signal. The signal is still contaminated by contributions from other components because the lateral separation cannot be reduced by squeezing. But the averaging along the measurement circle is so strong that for example for wave numbers above around $k_1 = 10^{-2}\,\mathrm{m}^{-1}$ less than half of the energy of the target spectrum is expected to be detected by the lidar.

First, when the model is compared with the measurement data, the chosen spectral tensor does not fit the actual wind conditions in the wave number range below $k_1 = 10^{-2}\,\mathrm{m}^{-1}$. The extra energy at low wave numbers compared to the spectral tensor model for this site has been observed before and is related to the inhomogeneous landscape at Høvsøre with its sea to land transition in the main wind direction (Sathe et al., 2015) and mesoscale effects that overlay the expected spectral gap (Larsén et al., 2016). Luckily, this does not severely impede the analysis since the most interesting effects are expected at higher wave numbers and tendencies can still be determined from the relative distances between the markers and lines without matching the absolute values. Next, the comparison of data from sonic measurements and VAD as well as SMC processed lidar data shows in the very low wave number range at $k_1 < 3 \times 10^{-3}\,\mathrm{m}^{-1}$ that VAD and SMC processing produce similar results with a slight tendency towards lower energy densities in the SMC measured spectrum that is not found in the model computations. A possible explanation is that the fluctuations of the $u$- and especially the $w$-component in the real wind field are not perfectly correlated, i.e. the frozen turbulence hypothesis which the model assumes is slightly violated. The result is a small contribution of $w_{wind}$ on $u_{lidar}$ that appears to a greater extent in the VAD processed spectrum. The reason for the difference is that the correlation is closer to unity in the case of SMC processing.

Apart from some exceptions (e.g., at $k_1 = 3 \times 10^{-3}\,\mathrm{m}^{-1}$), a relatively increasing averaging effect towards higher wave numbers is found for the lowest wave numbers as expected. In the wave number range $k_1 = 10^{-2}\,\mathrm{m}^{-1}$ to $6 \times 10^{-2}\,\mathrm{m}^{-1}$ the sonic spectrum and the VAD spectrum follow the corresponding modelled spectra nicely through the first resonance point. That shows that the cross-contamination caused by longitudinal separation is present in the measurements and is properly modelled.

The spectrum derived from SMC processed data shows a clear tendency towards its modelled spectrum but does not completely reach it. It does not show the resonance effect seen for VAD processing, but the overall energy level is higher than predicted for $k_1 > 10^{-2}\,\mathrm{m}^{-1}$. It is not possible to determine what causes this deviation. One possible reason is that the model assumes a perfect delay of the measurement timing. In reality this is not possible due to only discrete acquisition times being available. Also the air packages are in reality not always advected with the exact mean wind speed and direction. Both imperfections justify that the behavior of real SMC processing lies in between the modelled SMC and VAD processing.

For $k_1 > 7 \times 10^{-2}\,\mathrm{m}^{-1}$ VAD and SMC processed data are nearly identical. As shown in e.g., Schlipf et al. (2010), the assumption of frozen turbulence is not valid for high wave numbers. In this region, fluctuations separated by the distances between the relevant focus points are uncorrelated and the squeezing has no effect. The lack of coherence also explains that the values are higher than predicted because the $u$-spectrum is highly contaminated by $w$- and $v$-fluctuations.

### 6.1.2  2-beam processing

The plotted model spectrum for the conventional 2-beam processing method shows a significantly lower averaging effect compared to whole circle processing methods at all wave numbers except in the very low wave number region, where the methods are expected to perform similarly well.

With the 2-beam method it is expected that fluctuations with the highest wave numbers analyzed are to some extent included in the spectrum, while they were close to zero when circle processing was applied. The normal 2-beam processing in the model is prone to cross-contamination at both resonance points (vertical dashed lines). This situation is explained in detail in Sect. 2.5. In contrast, the method of squeezing applied to the 2-beam processing shows as expected no cross-contamination in the model calculations.

Overall, spectra calculated from the 2-beam processed measurement data show good agreement to the model. It is important to keep in mind that, due to the poor fit of the measured spectra of the horizontal wind components and the modeled spectra at low wave numbers, we can compare the relations between the different methods but not absolute values. At low wave numbers, the measured spectra are on average closer to the target spectrum than in the case of circle processing. The slightly lower energy content of "squeezed" measurements that we observed and explained for circle processing is found here as well. Also, when it comes to deviations from the modelled behavior like for example the higher energy density at some wave numbers (e.g., $k_1 = 3 \times 10^{-3}\,\mathrm{m}^{-1}$), we find similar tendencies as in circle processing, and the reason is likewise unclear.

The strong cross-contamination at the first resonance frequency is clearly represented in the normal 2-beam process-

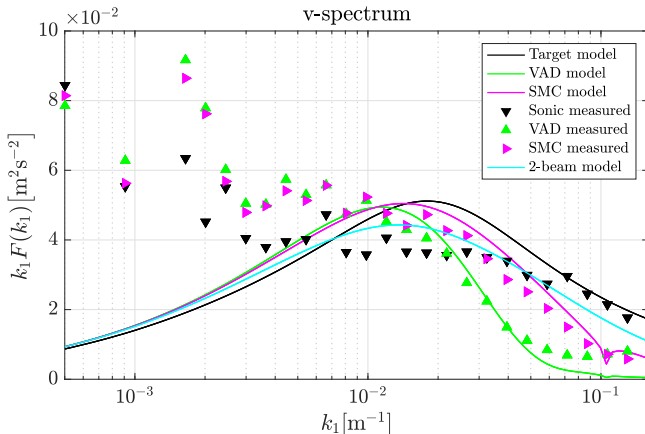

**Figure 6.** Modelled (solid lines) and measured (triangle markers) $v$-spectra from all data processing methods. Colors correspond to processing method.

ing and can be completely avoided by squeezing the two focus points to virtually one point. It is worth mentioning that the squeezing procedure works more like expected when applied to the 2-beam method than when applied to the circle processing. This can be explained by the error caused by not having continuous but only discrete delaying times $\tau$ available. The relative impact of this error is lower in the case of the 2-beam method because then the maximum separation distance $D_C$ must be compensated for. In circle processing mode, the shorter separations where the relative error is larger also contribute to the result.

At $k_1 > 7 \times 10^{-2}\,\mathrm{m}^{-1}$ the two processing methods result in nearly identical values again, and we assume the lack of coherence of short eddies to be the cause also here.

### 6.2 v-spectra

Figure 6 shows the spectra of the $v$-fluctuations for all available data processing methods from both measurement data (triangle markers) and the corresponding model predictions (solid lines). Also here, we first discuss the results from processing the whole measurement circle shown in Fig. 6a, followed by the discussion of the results of the 2-beam method shown in Fig. 6b.

#### 6.2.1 Circle processing

The modelled spectra of conventionally VAD processed lidar measurements predict energy densities that slightly exceed the target spectrum for very long fluctuations with $k_1 < 1.3 \times 10^{-2}\,\mathrm{m}^{-1}$. This behavior can be explained by uncorrelated $w$-fluctuations between the eastern and western sides of the measurement circle that contaminate the $v$-signal. This contamination is slightly stronger than averaging that is very weak at low wave numbers.

By contrast, fluctuations shorter than approximately $k_1 = 1.3 \times 10^{-2}\,\mathrm{m}^{-1}$ appear dampened in the spectrum, and fluctuations with higher wave numbers $k_1 > 10^{-1}\,\mathrm{m}^{-1}$ are not even present in the $v$-spectrum due to the strong averaging. Unlike the $u$-spectrum, the $v$-spectrum does not have characteristic behavior around the first resonance wave number. This is not surprising because the lines-of-sight that are the most important for the detection of $v$-fluctuations lie, according to Eq. (26), orthogonal to the mean wind direction in which turbulence is advected. So, no resonance occurs.

When the model spectrum for SMC processing is analyzed, we find a higher variance for all wave numbers above approximately $k_1 = 1.3 \times 10^{-2}\,\mathrm{m}^{-1}$. Reduced averaging along the measurement circle is the reason for the higher energy in the SMC spectrum. It is caused by the following: The process of squeezing reduces the longitudinal separation of the focus points ideally to zero while the lateral separation remains unchanged. We know that the lines-of-site perpendicular to the mean wind direction on both sides of the measurement circle are the most important for the determination of $v_{lidar}$. Let us assume these are the easterly and westerly beams. The exact east- and westbound beams are not affected by the process of squeezing. But for example the north-east and the south-east beam (respectively the north-west and north-east on the other side) see different turbulent structures in conventional VAD processing. With SMC processing, these two beams see the same structure. In the subsequent calculation of the $v$ component all lines-of-sight are combined and the pairs of radial velocities that lie in line with the mean wind contribute with the average of their amplitudes. This average of amplitudes is lower than the common amplitude measured by the beam pairs under SMC processing. More simply, there is less averaging along the measurement circle when SMC is applied. As a result, the spectrum of SMC shows higher energy densities for all wave numbers where uncorrelated fluctuations dominate.

Now we compare the measurements with the model. Unfortunately, similar to the $u$-fluctuations, the target spectrum does not represent the sonic measured values properly, especially for low wave numbers. We will therefore concentrate on the tendencies and proportions between the spectra from different methods. While the model predicts the behavior at the lowest wave numbers more or less satisfactorily, we are faced with two outliers at $k_1 = 1.65 \times 10^{-3}\,\mathrm{m}^{-1}$ and $k_1 = 2 \times 10^{-3}\,\mathrm{m}^{-1}$ where both the VAD and SMC processing lead to excessive energy estimations. The reason is unclear and not further investigated. At all other wave numbers, the agreement of model and measurements is very satisfactory. In particular, the differences between the two methods are found in the measurements, as predicted. The good agreement between model spectra and measurement spectra at wave numbers above approximately $k_1 = 2 \times 10^{-2}\,\mathrm{m}^{-1}$ might be surprising with regard to the poor agreement of sonic measurements and target spectrum. The reason is that the shape of the lidar $v$-spectra is mainly determined by the

cross-contamination from the $w$-component which, as we describe in 6.3, agrees better with its model representation.

The identity of VAD and SMC derived measurement spectra that we saw for $u$-fluctuations for $k_1 > 7 \times 10^{-2}\,\mathrm{m}^{-1}$ is found here at $k_1 > 10^{-1}\,\mathrm{m}^{-1}$. The reason is obvious when we look at the relevant longitudinal separation distances. They are much shorter when processing $v$-fluctuations than $u$-fluctuations, and the assumption of frozen turbulence is more valid for short separation distances. Therefore squeezing can maintain its effect into a somewhat higher wave number region.

### 6.2.2    2-beam processing

When the 2-beam method is applied, i.e., using only the east and west beam to derive the $v$-component of the wind vector, the method of squeezing has no effect. In comparison with the whole circle processing, the 2-beam method is characterized by lower energy estimates at low wave numbers and higher energy estimates at higher wave numbers (see Fig. 6). One reason for the first is assumed to be the lower coherence of $v$-fluctuations separated by the full distance $D_C$. That implies that 2-beam processing gets a somewhat lower contribution of $v_{wind}$ to $v_{lidar}$. A second reason is that there is not cross-contamination from $u$ on $v$ occurring for the 2-beam processing. The higher energy content at high wave numbers results from the absence of averaging along the measurement circle.

The model cannot be compared with measurements because the line-of-sight velocities of the east and west beams were erroneous. The absolute values we measured are unrealistically biased towards non-zero values. This effect has been previously reported (Mann et al., 2010; Dellwik et al., 2010). We included the model behavior of 2-beam processing for the sake of completeness and to show that the availability of reliable measurement data for the east and west beams would be of hardly any use.

### 6.3    w-spectra

Figure 7 shows the spectra of the $w$-fluctuations for all processing methods from both measurement data and the corresponding model predictions. Again, we discuss the results from processing the whole measurement circle first and then the results of the 2-beam method.

### 6.3.1    Circle processing

To begin with, we compare the model predictions of conventional VAD processing and the new SMC method against one another and with regards to the $w$ target spectrum. The results of the actual measurements follow. The model prediction of the conventionally processed VAD lidar data shows some attenuation of the spectral energy even for very low wave numbers. The reason is mainly the infinitely long tails of the line-of-sight averaging function and to a lesser extent

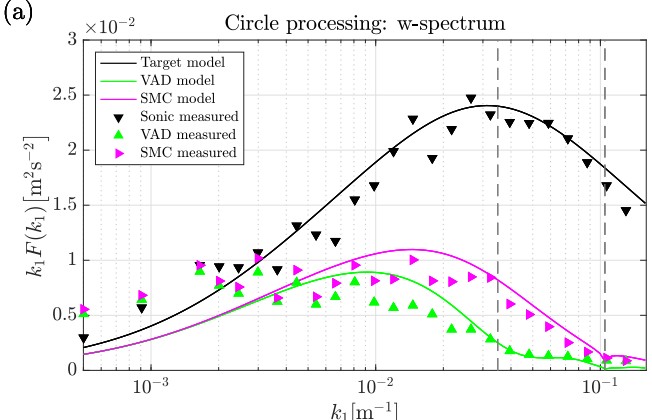

(a)

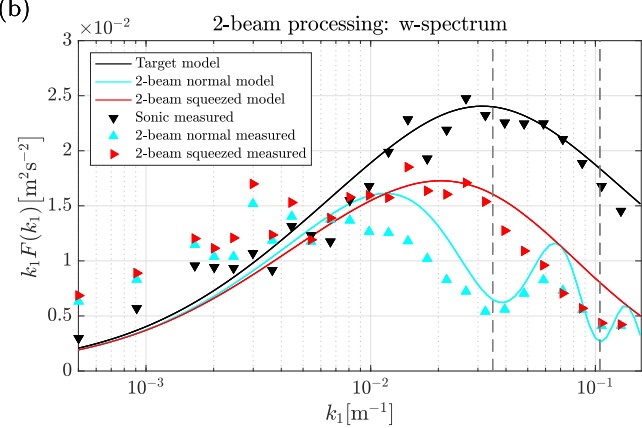

(b)

**Figure 7.** Modelled (solid lines) and measured (triangle markers) $w$-spectra from data processing where (a) all radial measurements are used and (b) only two beams are used. Colors correspond to processing method. The grey vertical dashed lines represent the first and second resonance wave number.

the averaging along the measurement circle. Both averaging effects become quickly stronger for increasing wave numbers. The spectrum from VAD processed data is expected to drop at the first resonance point marked with a grey dashed vertical line in Fig. 7. This drop is minor due to the overall low energy level present in the spectrum. The spectrum reaches a value near its final minimum with variance values close to zero already at around $k_1 = 5 \times 10^{-2}\,\mathrm{m}^{-1}$ just after crossing the first resonance point. $w$-fluctuations with higher wave numbers are not detectable with conventional VAD processing. According to the model, the SMC processing improves the situation slightly by removing the longitudinal separation that makes lidar blind to $w$-fluctuations at the resonance points with VAD processing. Squeezing the measurements also helps improve the measurements well above and below the resonance wave number. But still, due to the remaining averaging effects, only a minor fraction of the energy in the vertical wind can be detected with both methods at wave numbers above roughly $k_1 = 10^{-2}\,\mathrm{m}^{-1}$.

Unlike for the $u$ and $v$-component, the fit between target spectrum and measurement data is good for the $w$-component also in the low wave number region. The results of Larsén et al. (2016) show that the spectra for vertical fluctuations are not prone to contributions from the mesoscale spectrum. The measurement data overall support these model predictions and show that the process of squeezing functions well over the entire frequency range in this study. In detail, we only find some mismatch for very low wave numbers where $k_1 < 10^{-3}\,\mathrm{m}^{-1}$. The measured spectra lie above the target spectrum here although we expected some attenuation. The discrepancy is caused by the real $u$-wind spectrum being much higher than the underlying target spectrum, see Fig. 5. We already found that large scale $u$-fluctuations are also not perfectly correlated and thus contaminate the measured lidar spectra, which is not considered in the model. At higher wave numbers we find reasonable forecasting of measured $w$-spectra by the model.

### 6.3.2 2-beam processing

The modelled 2-beam spectra in Fig. 7b lie considerably closer to the target $w$-spectrum for all wave numbers. That can be explained by the absence of circle averaging. The strong influence of resonance visible at the two first resonance wave numbers underlines the importance of squeezing when striving for more realistic spectra from lidar measurements.

At low wave numbers with $k_1 < 10^{-2}\,\mathrm{m}^{-1}$ the measured spectra contain higher energy densities than modelled. A similar but less pronounced effect was found in circle processing only at the lowest wave numbers. The explanation we gave there must therefore be supplemented by mentioning that the assumed decorrelation is stronger for the maximal separations that are involved in the 2-beam method. The further comparison of spectra from experiment and model show that the process of squeezing also leads to the expected effect in the case of using only two beams to determine the $w$ component of the wind vector. As in the case of $u$-fluctuations, this statement must be limited to wave numbers $k_1 < 7 \times 10^{-2}\,\mathrm{m}^{-1}$.

### 6.4 Extended discussion

The results discussed here are extracted from a single data set that covers one measurement height and a narrow band of mean wind speeds, turbulence conditions and inflow directions at a single location. The reason for working with such a limited data set lies in the fact that very little data is available where a commercial VAD scanning wind lidar, collocated to a meteorological mast, is scanning continuously at one height level, while saving at least the line-of-sight velocities. Currently, the only option to save line-of-sight velocities acquired by a ZephIR 300 is to stream the data manually to a connected PC. The situation gets further complicated by the fact that in the normal "profiling mode" the lidar focuses to a reference height of 38m periodically for filtering purposes. So, the only known way to focus at one particular altitude continuously is to switch the unit to "turbine mode". In this way, we acquired some data for the investigation, but their overall quality was lower than the historic data that we eventually selected as the best available data.

In further studies different set-ups and turbulence conditions should be investigated. Changing the measurement height has the strongest influence on the lidar-derived spectra. For example, increasing the measurement height would, first, make the averaging along the measurement circle more severe due to the increased measurement circle diameter. Second, the resonance wave numbers are then shifted towards lower values, which leads to different cross-contamination due to lateral separation. Third, the cross-contamination due to lateral separation becomes even more severe due to the longer separation distances of opposite line-of-sight beams. Fourth, a further increase of the focus distance leads to even stronger line-of-sight averaging. Fifth, the time lag that is introduced for squeezing must be longer, and the frozen turbulence hypothesis loses some more of its validity. Changing the half cone opening angle to a smaller value would on the one hand reduce the first three of the aforementioned effects effectively, but on the other hand it would lead to much stronger cross-contamination due to the increased sensitivity for $w$-fluctuations according to Eqs.14 and 16. Lidar measurements at lower mean wind speeds give the turbulence more time to evolve while crossing the measurement circle, which might lead to a deviation from the predicted spectra at somewhat lower wave numbers than observed in our results. The numerical models will work for all turbulence intensities, and the shape of the spectra is mainly determined by the degree of anisotropy and the turbulence length scale. Atmospheric stability conditions other than neutral would not change the way the lidar measures. But a modified spectral tensor model like the one presented in Chougule et al. (2017) could be used to better compare model values with experimental results.

## 7 Conclusions

This paper presents two advanced data processing methods for improving turbulence spectrum estimations with VAD scanning wind lidars, with an aim to reduce cross-contamination and averaging effects. The models of these approaches, developed in Sect. 4, are supported by the comparison with experimental data. Discrepancies can be explained for the most part by the limitations of the frozen turbulence hypothesis that underlies the model calculations yet has slightly reduced validity in real measurements. Also the fact that the spectra in the experiment do not agree very well with the spectral tensor model is a cause of differences.

We found that the method of squeezing eliminates the resonance effect caused by the longitudinal separation of combined measurement points successfully. It also considerably reduces the averaging along the measurement circle.

The method of using only two beams for the estimation of the $u$ and $w$ components of the wind vector eliminates the averaging along the measurement circle completely. When it is combined with the method of squeezing, the measurements deviate from the sonic measurements mainly due to line-of-sight averaging. This combination of both methods substantially improves the measurability of the $w$-spectrum, which is hardly measurable with current VAD processing.

Accurate measurements of the $v$-spectrum remain difficult, even with the approaches described here. The 2-beam method is not applicable to current continuous-wave lidars which in most cases are homodyne. Whether the use of squeezed measurement circles always leads to systematically better results is unclear, because the resulting spectra are dominated by contamination from w-fluctuations of the wind.

In conventionally processed lidar data, cross-contamination compensates for averaging effects, meaning that in general, total variance might be close to target values, but for the wrong reasons. For systematically better turbulence measurements from VAD scanning lidars, the findings presented here should be included in raw data processing. Both approaches presented here can be applied to any existing VAD scanning continuous-wave profiling lidar unit.

*Code and data availability.* Inquiries about and requests for access to data and source codes used for the analysis in this study should be directed to the authors.

*Competing interests.* The authors declare no competing interests.

*Acknowledgements.* This research project was supported by Energy and Sensor Systems (ENERSENSE) at the Norwegian University of Science and Technology.

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
