# Peer review of "Better turbulence spectra from VAD scanning wind lidar"

_Atmospheric Measurement Techniques, 2018_

## Referee Comment (RC1) · Anonymous Referee #1 · 14 Jan 2019

General Comments:

The manuscript describes two approaches to reduce the effects of averaging and cross-correlation between the different velocity components that typically compromise turbulence measurements from lidars and complicate the comparability of those measurements with in-situ point measurements, e.g. performed by sonic anemometers. The topic is highly relevant under the aspect of the potential and limitations to derive reliable atmospheric turbulence data from lidar remote sensing and falls clearly into the scope of AMT. The manuscript is clearly structured and and gives a thorough description of the theoretical background and the mathematical formulation of the models to simulate the different lidar scanning and analysis methods. It is, however, not always easy to read. The introduction would be strengthened by some discussions on the im-

plications of the performed study on practical applications e.g. for future boundary layer research in general, and for wind energy applications, as e.g. related to the investigation of loads of wind turbines, in particular. The main weak point of the study is the very limited data set used for the comparison between measurements and simulations (only about 5 consecutive hours during one day), representing one single situation of wind conditions. In terms of wind energy, this wind speed is clearly above the rated wind speed of typical wind turbines, with the blades already considerably pitched and thus not so prone to turbulent loads. For corresponding investigations and analyzes it would be highly interesting how the method works around and below rated wind speed, e.g. for 12 and 8 m/s. Investigating more situations might also enlighten further on the large discrepancies (in the same magnitude as discussed improvements due to the new proposed processing methods) between target spectrum and sonic measurements for u and v. As long as this is not sufficiently understood, I would also doubt the significance of many of the presented results.

Specific comments:

1) Title: "Better" in the title immediately associates with "than what"; I would at least think about changing to "Improved", if not extending the title a bit more to make clearer what the reader can expect.

2) P1, L20: the actual manufacturer/distributor should be mentioned here

3) P2, L14 (and other instances): "however" should usually be separated by comma on both sides

4) P2, L24-25: what is a "correlation twice as strong", please quantify in more detail; in addition is the fact "that the correlation increases when a time shift related to the mean wind is taken into account" presented as surprising, but it is of course not and a very standard method in many measurement applications (e.g. to correct for wind speed and direction dependent time shifts between dislocated sensors, e.g. sonic anemometers and infrared gas analyzers for the determination of latent heat and $CO_2$

fluxes)

5) P3, Figure 1: unconventional coordinate system, in particular for meteorology, where usually u defines the E-W direction and v the N-S direction

6) P4, L14-15: The quantitative results of those other experiments should be shortly presented with the corresponding references; I also miss a reference to one of the basic statements on the issue by Willis and Deardorff, 1976 (ïĄşU/U < 0.5)

7) P5, L17: maybe better "in contrast" instead of "by contrast"

8) Section 2.4.1: would be nice to quantify and present in a table some of the key parameters for the used instrument, e.g. IR , df and a0

9) P6, L18-19: "... we cannot distribute the lost small scale fluctuations on spectral frequencies.....", but are you able to quantify the overall amount/importance?

10) P6, L29: insert "," after "composed of"

11) P8, Figure 2: some of the labels are by far too small and not or only very difficult to read; please improve

12) P7, L12: "This leads to a redistribution of energy among the velocity components u, v, and w." If it is a pure redistribution this should mean that TKE spectra derived from the measurements should be unaffected, correct? Gives this a tool for a potential quality assurance of the lidar derived variances?

13) P12, Figure 4: please specify in the caption altitude level and average wind speed for the presented example

14) P12, L4: insert blank after reference Mann et al. (2010)

15) P13, L4-6: can you quantify/reflect upon the order of magnitude of the uncertainty that is introduced by this assumption

16) P14, Equation 23: should it read sin2c instead of sinc2 ?

17) P17, L18-21 (measurement description): very superficial description here, I suggest to include a few key infos in addition to the Pena et al. reference; what was the measurement frequency of the sonic; is it ensured that the measurements are unaffected by flow distortion/mast effects during the investigated case

18) P18, L5: "The data rate is thus for all methods 1 Hz"; does this also include the sonic data? If so, how are the sonic 1 Hz data sampled? I assume the sonics run somewhat between 10 and 50 Hz, so you could create 1 Hz data either by averaging your raw data, or picking one raw value every s. The corresponding selection will of course have an influence on your final spectra.

19) P19, Figure 5 (also Figures 6 and 7): I suggest to reconsider your presentation of the results ; in particular the black and blue squares are very hard to distinguish; this makes it very hard to follow the discussion of the results; a quick fix could be to consider in addition to the different colors also different symbols (e.g. square, star, triangle)

20) P19, Figure 5: why is the target model so far away from the sonic measurements; are there any effects of flow distortion visible (connects to cmment 17)

21) P21, L16: ".... show very good agreement to the model"; I feel this is a very strong statement; I can support this only for kl>10-2, but not for the rest

22) P23, L14-16: the unfortunate mis-representation of the sonic measurements by the target spectrum is again a great concern; how confident can you be in the discussion of improvement caused by your new processing methods, when the discrepancies between the target model and the sonic measurements are of the same magnitude or even larger; Again I would strongly recommend to also look at and present results for different synoptic/wind situations to provide some evidence where this mis-match between target spectrum and measurements originates from.

23) P 24, Figure 7: its puzzling that the sonic measurements and the target model fit

very well in the case of w, while the match it is rather poor for u and v;

24) P26, L6-7; This is a crucial issue of the manuscript and has to be elaborated in more detail (see also my corresponding concern in the general comment section)

25) Formatting of references is inconsistent a. Journal names abbreviated/not abbreviated b. DOI given or not c. Incomplete, pages missing: e.g. Newman et al., 2016; Pena et al., 2015; Sathe and Mann, 2012;
* * *

---

## Referee Comment (RC2) · Anonymous Referee #2 · 23 Jan 2019

Within this manuscript, the authors present a new method to calculate the turbulence spectra from measurements from a CW Doppler lidar using velocity-azimuth display (VAD) scans. The authors use both a modelling and measurement approach, comparing lidar and 'truth' sonic anemometer measurements, to demonstrate that their proposed method provides more accurate turbulence spectra than existing techniques from VAD scans.

The novel subject matter is timely and of great interest to the readership of AMT. However, much of these results relies on only 5 hours of data under a narrow set of conditions, which really has limited the significance of the present study. Additionally, the modeled results are not in good agreement with the observations, and these differences are hardly explained or investigated. This is concerning, given the significant

role of the modelling results in this study. With these issues, I recommend the article be reconsidered for acceptance in AMT after major revisions.

Major comments:

a) Figs. 5, 6, 7 and throughout Sect. 6: In these plots (especially 5, 6), there is a large disagreement between the modeled spectra and the observed spectra, particularly at low wavenumbers. This is concerning given that much of the presented results rely on the accuracy of the model, and it appears that the model is not accurately representing real lidar measurements. The only reason given for this difference between the real and modeled spectra is that it is a result of the heterogeneous landscape (p. 20, line 21). Given the importance of the modeled spectra in this study, this justification is insufficient and it was hardly discussed in the cited reference. The authors must investigate these differences herein further to understand their root cause, otherwise the use of the modeled spectra herein is suspect.

b) This manuscript in particular would benefit greatly from a 'Discussion' section between the results and conclusions that would link the results here to possible wider adoption across a variety of locations/seasons/times. This is especially important for this study given the fact that its results are solely from 5 hours of lidar data under high-wind conditions during the daytime in winter. The authors should discuss how the results are expected to vary under different conditions (weaker winds, very stable/unstable, etc), for measurements at different altitudes, circle diameters, half opening angles, or anything else the authors think would be relevant to any user that would try to apply this method elsewhere.

Alternatively, instead of adding a discussion section the authors could expand their study to more time periods under different atmospheric conditions.

Specific comments:

a) P. 2 line 12: Somewhere around here it would be appropriate to reference Eber-

hard et al (1989) as one of the first studies where Doppler lidars were used to profile turbulence using VADs.

b) Figure 2: Parts of this figure (especially the subfigure showing the lidar beams) are extremely small and difficult to read. This should be improved, making the figure larger would help. I also suggest adding to the caption describing how the top plot visualizes the lidar beam positions.

c) P. 8 lines 5-13 & Fig. 3: This should be moved to later in the paper, perhaps Sect. 4. At this point, the reader has no context to understand the details of what is being shown as the model has not been described.

d) Fig. 3 (and throughout): The units for power spectra in the atmospheric science community are generally mˆ-1, not rad mˆ-1. These units appear throughout the text, in the table, and figure.

e) Eq. 10 & 11: Be consistent with these equations. Eq. 11 gives the entire variance for v while Eq. 10 only gives the variance contamination for u, but the subscript notation on the left-hand side for both indicates total variance. Also, Eq. 11 does not appear to be derived correctly (and is inconsistent with Eq. 10). Why is the $\frac{1}{2}$ factor in the equation? The logic for how these Eqs are derived is not obvious (should be clarified), but should there also be a term for the covariance (u'w' and v'w' overbars) on the right-hand side?

f) P. 9 lines 8-16, 20-21: This text should also be moved to Sect. 4 where the readers will have the appropriate context to understand the discussion here. The rest of the text after Eq. 11 and before Sect. 3 can be combined into one paragraph.

g) Fig. 4: Nice figure. It could be improved if you added a reference vector for the mean wind to show how the field is advected. It would also be beneficial in the lower plot to highlight in a different color which pairs of measurements are used in the two-beam method.

h) Sect 4: Please make sure to explain and define all variables and notation. In particular, I could not find definitions for: e_1, Φ, and T. The notation of _z was also not described.

i) Sect 5.1: Add a sentence here cross-referencing table 1 to summarize the experiment. Please also expand this discussion. The time period is 5 hours, it is unlikely the wind speed was constant that entire time. How much did it vary? How much did the turbulence intensity vary? What was the stability of the boundary-layer? Given the location, I expect near-neutral stability, but it would be good to verify and quantify the stability. How were the resonance values in Table 1 determined? What was the precision of the lidar measurements?

j) P. 19 line 2: Please add a more throughout description of how these spectra are made. Is this an average of the individual 10-min spectra? Were outliers in the spectra removed in making this plot? How much did the actual spectra vary of the time period? The description is insufficient.

k) P. 19 line 8: What is the 'target spectrum'? Is this simply the modeled spectrum assuming certain characteristics of the flow garnered from the sonic anemometer measurements?

l) Fig. 5: These two plots can be combined as the axes are identical and much of the data overlaps. By combining the plots, the VAD/SMC and two-beam methods can also be compared. Also state in the caption what the vertical dashed lines indicate.

m) P. 20 line 25: By this, do you mean that the frozen turbulence hypothesis is not completely valid as the wind field evolves in time as it advects through the measurement volume? If so, please clarify that here.

n) P. 20 line 34: Could this deviation also be caused by random errors in the lidar measurements resulting in a noise floor above the modeled value (related to last point in i) above)? Based on the model description in Sect. 4, measurements are modeled as precise (without any random error).

o) Fig. 6: These two subplots can be combined (if kept, see comment p), as b) only contains one additional piece of information (red line) that could be easily overlaid on a) for comparison.

p) Sect. 6.2.2: This section and Fig. 6 b can be removed. If this method is not even applicable to real CW lidar measurements due to the ambiguity around a 0 Doppler velocity, why even present it as a method?

q) P. 25 line 20: Recommend changing the term from 'very good' to 'reasonable'. There are still non-trivial differences between the modeled and observed spectra in this reviewer's opinion.

Editorial corrections

a) P. 1 Line 19: Add hyphen to Velocity-azimuth display.

b) P. 6 line 30: Reword to: 'Turbulence with a length scale below . . .'

c) P. 7 line 1: Remove 'to sense them'

d) p.7 line 10: Remove 'or cross talk'

e) P. 13 line 4: 'is' should be 'are'

f) Eq. 23: Should be 'sin' instead of 'sinc'.

g) P. 18 line 2: Remove 'used'

References:

Eberhard, W.L., Cupp, R.E. and Healy, K.R., 1989. Doppler lidar measurement of profiles of turbulence and momentum flux. Journal of Atmospheric and Oceanic Technology, 6(5), pp.809-819.

---

## Author Comment (AC1) · 5 Feb 2019

We thank Anonymous Referee #1 for providing their review of our manuscript. The attached .zip archive contains our response and, in a separate file, a latexdiff version of our revised manuscript.

Please also note the supplement to this comment: https://www.atmos-meas-tech-discuss.net/amt-2018-410/amt-2018-410-AC1-supplement.zip

---

## Author Response (AR1)

We thank the Referee #1 for the time and effort involved in reviewing our manuscript. We appreciate the very constructive and useful input a lot. See below for our point-by-point response.

*The manuscript describes two approaches to reduce the effects of averaging and cross-correlation between the different velocity components that typically compromise turbulence measurements from lidars and complicate the comparability of those measurements with in-situ point measurements, e.g. performed by sonic anemometers.*
*The topic is highly relevant under the aspect of the potential and limitations to derive reliable atmospheric turbulence data from lidar remote sensing and falls clearly into the scope of AMT. The manuscript is clearly structured and and gives a thorough description of the theoretical background and the mathematical formulation of the models to simulate the different lidar scanning and analysis methods. It is, however, not always easy to read. The introduction would be strengthened by some discussions on the implications of the performed study on practical applications e.g. for future boundary layer research in general, and for wind energy applications, as e.g. related to the investigation of loads of wind turbines, in particular. The main weak point of the study is the very limited data set used for the comparison between measurements and simulations (only about 5 consecutive hours during one day), representing one single situation of wind conditions. In terms of wind energy, this wind speed is clearly above the rated wind speed of typical wind turbines, with the blades already considerably pitched and thus not so prone to turbulent loads. For corresponding investigations and analyzes it would be highly interesting how the method works around and below rated wind speed, e.g. for 12 and 8 m/s. Investigating more situations might also enlighten further on the large discrepancies (in the same magnitude as discussed improvements due to the new proposed processing methods) between target spectrum and sonic measurements for u and v. As long as this is not sufficiently understood, I would also doubt the significance of many of the presented results.*

We agree with the referee's helpful suggestion and have added some short discussion on the implications of improved lidar derived turbulence spectra to the introduction.
The referee considers the very limited data set we use for the comparison between measurements and simulations as the main weak point of our manuscript. We agree that an investigation with more data would have been beneficial. The reason for working with this limited data set lies in the fact that very little data are available where a commercial VAD scanning wind lidar, collocated to a meteorological mast, is scanning continuously at one height level, while saving at least the line-of-sight velocities. Currently, the only option to save line-of-sight velocities acquired by a ZephIR 300 is to stream the data manually to a connected PC. The situation gets further complicated by the fact that in the normal "profiling mode" the lidar focuses to 38m periodically for filtering purposes. So, the only known way to focus at one particular altitude continuously is to switch the unit to "turbine mode". In this way, we acquired some data for the investigation, but their overall quality was lower than the historic data that we eventually selected as the best available data.
This data set shows turbulence spectra that strongly deviate from the shape of spectra derived from the Mann spectral tensor in the low wave number region for both horizontal velocity components. The vertical component fits well. Larsén et al. (2016) are confronted with a similar situation while they investigate spectra based on wind data of complete years for the same location, Høvsøre. The understanding of this behavior is that mesoscale effects are visible in the spectra which are not considered by the turbulence model. We can therefore not necessarily expect to reach a good fit between model spectra and sonic measurements by simply using different or more data. Instead, we based our discussion on relative spectral energy distributions

between the different data processing methods and compared them with the corresponding model predictions. In this way, we did not depend on a fit of the absolute values. This situation is unfortunate but not alarming. In particular, the effects of the two introduced methods are found mainly in the higher wave number region where the fit is at least better.

In particular, the referee raises a concern about the high mean wind speed during the measurements. Such a high mean wind speed is clearly far above average, but for the turbulence spectra it is of no importance if the wind speed is above or below rated.

Applying the method of "squeezing" to data acquired by the DBS scanning wind lidar Windcube is the object of another ongoing investigation. In that case, we have a huge amount of suitable data available and might draw conclusions with respect to different wind directions, measurement heights, atmospheric stability conditions and so on.

Here, we do not intend to investigate specific turbulence situations but want to show that the ideas behind the two novel methods work in practice and lead to the results that can be predicted by the numerical models we present. We also see and point out the limitations of the methods. Thus, we believe that the use of more or different data would not lead to different conclusions and is therefore not essential.

Larsén, X. G., Larsen, S. E., and Petersen, E. L.: Full-Scale Spectrum of Boundary-Layer Winds, Bound. Lay. Meteorol., 159, pp. 349–371, https://doi.org/10.1007/s10546-016-0129-x, 2016.

*Specific comments:*

*1) Title: "Better" in the title immediately associates with "than what"; I would at least think about changing to "Improved", if not extending the title a bit more to make clearer what the reader can expect.*

      The word "Improved" was part of a preliminary title. Eventually, we decided for the word "better" as the simpler of the two synonyms. We actively decided against a more detailed title because both data processing methods are new and as yet unknown. Giving for example their names would, unfortunately, not be sufficient to explain what we did.

*2) P1, L20: the actual manufacturer/distributor should be mentioned here*
      The name of the manufacturer "Zephir Ltd. / ZX Lidars" was added.

*3) P2, L14 (and other instances): "however" should usually be separated by comma on both sides*
      The missing commas were added.

*4) P2, L24-25: what is a "correlation twice as strong", please quantify in more detail; in addition is the fact "that the correlation increases when a time shift related to the mean wind is taken into account" presented as surprising, but it is of course not and a very standard method in many measurement applications (e.g. to correct for wind speed and direction dependent time shifts between dislocated sensors, e.g. sonic anemometers and infrared gas analyzers for the determination of latent heat and CO2 fluxes)*

      The increased correlation is of course not surprising, and it was indeed the motivation for the squeezing method. Although, as the referee mentions, retarding signals is used in many different applications, it is new in the field of profiling wind lidar. Bardal and Sætran (2016) give an indication of the partial validity of the assumption of frozen turbulence for wind at scales that are relevant for our work. We added some information about the setup and described the results of Bardal and Sætran (2016) in more detail.

*5) P3, Figure 1: unconventional coordinate system, in particular for meteorology, where usually u defines the E-W direction and v the N-S direction*

Since $u$ is defined as the wind velocity in the mean wind direction, $u$ can also be aligned with the E-W direction. We agree that the left-handed coordinate system is confusing when the reader expects the one that is usually used in meteorology. We want the coordinate system to be the same as in Sathe and Mann (2012) and other publications so that the model for DBS scanning given there and the one we present here for VAD scanning use the same notation and signs.

*6) P4, L14-15: The quantitative results of those other experiments should be shortly presented with the corresponding references; I also miss a reference to one of the basic statements on the issue by Willis and Deardorff, 1976 (ï ̧A ̧sU/U < 0.5)*

That is true. We included the statement of Willis and Deardorff, 1976 and added the qualitative results of Schlipf et al., 2010.

*7) P5, L17: maybe better "in contrast" instead of "by contrast"*

We changed the formulation according to the referee's suggestion.

*8) Section 2.4.1: would be nice to quantify and present in a table some of the key parameters for the used instrument, e.g. lR, df and a0*

A reference to Table 1 that gives these values is now added in the text.

*9) P6, L18-19: "... we cannot distribute the lost small scale fluctuations on spectral frequencies...", but are you able to quantify the overall amount/importance?*

The formulation is now different: "The effect of line-of-sight averaging is considered in the numerical models and the discussion in this study. But none of the presented data processing methods can avoid the line-of-sight averaging effect."

*10) P6, L29: insert "," after "composed of"*

Thanks, a comma is inserted.

*11) P8, Figure 2: some of the labels are by far too small and not or only very difficult to read; please improve*

Figure 2 has been improved. We substituted the tiny label "Lidar" by a yellow symbol and updated the caption. All remaining labels are now according to the font size of the text.

*12) P7, L12: "This leads to a redistribution of energy among the velocity components u, v, and w." If it is a pure redistribution this should mean that TKE spectra derived from the measurements should be unaffected, correct? Gives this a tool for a potential quality assurance of the lidar derived variances?*

Good idea, but TKE spectra are not unaffected by the cross-contamination since the half cone opening angle is not equal but <45 deg. Thus, we are faced with a higher sensitivity for vertical fluctuations than for horizontal fluctuations. We do not know how much energy will be redistributed and can therefore not estimate if the TKE increases, decreases or if it stays more or less constant by the application of the squeezing method. All three cases might occur depending on the degree of anisotropy of the prevailing wind.

*13) P12, Figure 4: please specify in the caption altitude level and average wind speed for the presented example*

The caption now contains the information on $U$, $D_C$ and $f_S$.

*14) P12, L4: insert blank after reference Mann et al. (2010)*
        Done.

*15) P13, L4-6: can you quantify/reflect upon the order of magnitude of the uncertainty that is introduced by this assumption*
        If the assumption was violated, we would need to look at all the particular beam combinations as in e.g. for the pulsed lidar modelling in Sathe and Mann (2012). This would lead to very complicated calculations. As a result, we would see cross-contamination only for resonance frequencies that are multiples of $f_s$. We added to the text that "It is difficult to assess the magnitude of the error committed by the assumption of continuous measurements, but we assume it is negligible."

*16) P14, Equation 23: should it read sin2c instead of sinc2 ?*
        Eq. 23 contains the cardinal sine function *sinc* that not everyone might be familiar with. We therefore inserted an explanation into the text.

*17) P17, L18-21 (measurement description):  very superficial description here, I suggest to include a few key infos in addition to the Pena et al.  reference; what was the*
*measurement frequency of the sonic; is it ensured that the measurements are unaffected by flow distortion/mast effects during the investigated case*
        We included more information about the measurement setup, and we have now explained why we do not expect flow distortion in the selected measurement data. The measurement frequency of the sonic anemometers was 20 Hz. This information is added to section 5.2.

*18) P18, L5:  "The data rate is thus for all methods 1 Hz"; does this also include the sonic data?  If so, how are the sonic 1 Hz data sampled?  I assume the sonics run somewhat between 10 and 50 Hz, so you could create 1 Hz data either by averaging*
*your raw data, or picking one raw value every s. The corresponding selection will of course have an influence on your final spectra.*
        Good point. The following information is added: "The sonic anemometer measures with a rate of 20 Hz. This high frequency data is down-sampled by the use of the MATLAB function 'resample' to a frequency of 1 Hz. The function includes a low-pass filter to avoid anti-aliasing."

*19) P19, Figure 5 (also Figures 6 and 7):  I suggest to reconsider your presentation of the results ;  in particular the black and blue squares are very hard to distinguish;*
*this makes it very hard to follow the discussion of the results; a quick fix could be to consider in addition to the different colors also different symbols (e.g. square, star, triangle)*
        We appreciate the insight that some of the results might be difficult to read. For a better distinction of the results presented in the plots, we have replaced the square markers by triangles with different orientations. We also replaced blue lines and markers by cyan colored ones.

*20) P19, Figure 5: why is the target model so far away from the sonic measurements; are there any effects of flow distortion visible (connects to comment 17)*
        Analyzing the definite cause for the discrepancy between model and reference lies unfortunately out of the scope of this study, but we have extended our explanation of it and added a second, better reference: "The extra energy at low wave numbers compared to the spectral tensor model for this site has been observed before and is related to the inhomogeneous

landscape at Høvsøre with its sea to land transition in the main wind direction (Sathe et al., 2015) and mesoscale effects that overlay the expected spectral gap (Larsén et al., 2016)."

*21) P21, L16: "... show very good agreement to the model"; I feel this is a very strong statement; I can support this only for kl>10-2, but not for the rest*

       We changed our statement to: "Overall, spectra calculated from the 2-beam processed measurement data show good agreement to the model." And added the following: "It is important to keep in mind that, due to the poor fit of the measured spectra of the horizontal wind components and the modeled spectra at low wave numbers, we can compare the relations between the different methods but not absolute values."

*22) P23, L14-16: the unfortunate mis-representation of the sonic measurements by the target spectrum is again a great concern; how confident can you be in the discussion of improvement caused by your new processing methods, when the discrepancies between the target model and the sonic measurements are of the same magnitude or even larger; Again I would strongly recommend to also look at and present results for different synoptic/wind situations to provide some evidence where this mis-match between target spectrum and measurements originates from.*

       We are well aware that our statements about effects we see in the spectra have a thin data basis compared to other investigations. And we appreciate getting the feedback that this creates doubt about the overall results. To respond we would like to point out that our analysis is also based on the theoretical considerations we elaborated in sections 2 and 3. Most of the results are as expected and described as thoroughly as possible. Only minor effects cannot be explained satisfactorily, which is then mentioned in the text. The good coherence between what we know about the theory, the model predictions and the measurement results gives us confidence in the points we raise in our discussion.

*23) P 24, Figure 7: its puzzling that the sonic measurements and the target model fit very well in the case of w, while the match it is rather poor for u and v;*

       An explanation for the better fit of the model spectrum to the measurements in *w* than in *u* and *v* is given in Larsén et al. (2016). We added: "The results of Larsén et al. (2016) show that the spectra for vertical fluctuations are not prone to contributions from the mesoscale spectrum."

*24) P26, L6-7; This is a crucial issue of the manuscript and has to be elaborated in more detail (see also my corresponding concern in the general comment section)*

       See our comment in the general section prior to point 1).

*25) Formatting of references is inconsistent a. Journal names abbreviated/not abbreviated b. DOI given or not c. Incomplete, pages missing: e.g. Newman et al., 2016;*
*Pena et al., 2015; Sathe and Mann, 2012;*

       We thank the referee for mentioning the inconsistency in the list of references. The formatting is consistent now: journal names abbreviated, DOIs given wherever available and pages added wherever applicable.

We thank the Referee #2 a lot for a very useful and constructive review of our manuscript and appreciate the time and effort involved in it. We have provided a point-by-point response below.

*Within this manuscript, the authors present a new method to calculate the turbulence spectra from measurements from a CW Doppler lidar using velocity-azimuth display (VAD) scans. The authors use both a modelling and measurement approach, comparing lidar and 'truth' sonic anemometer measurements, to demonstrate that their proposed method provides more accurate turbulence spectra than existing techniques from VAD scans.*
*The novel subject matter is timely and of great interest to the readership of AMT. However, much of these results relies on only 5 hours of data under a narrow set of conditions, which really has limited the significance of the present study. Additionally, the modeled results are not in good agreement with the observations, and these differences are hardly explained or investigated. This is concerning, given the significant role of the modelling results in this study. With these issues, I recommend the article be reconsidered for acceptance in AMT after major revisions.*

*Major comments:*
*a) Figs. 5, 6, 7 and throughout Sect. 6: In these plots (especially 5, 6), there is a large disagreement between the modeled spectra and the observed spectra, particularly at low wavenumbers. This is concerning given that much of the presented results rely on the accuracy of the model, and it appears that the model is not accurately representing real lidar measurements. The only reason given for this difference between the real and modeled spectra is that it is a result of the heterogeneous landscape (p. 20, line 21). Given the importance of the modeled spectra in this study, this justification is insufficient and it was hardly discussed in the cited reference. The authors must investigate these differences herein further to understand their root cause, otherwise the use of the modeled spectra herein is suspect.*

*b) This manuscript in particular would benefit greatly from a 'Discussion' section between the results and conclusions that would link the results here to possible wider adoption across a variety of locations/seasons/times. This is especially important for this study given the fact that its results are solely from 5 hours of lidar data under high-wind conditions during the daytime in winter. The authors should discuss how the results are expected to vary under different conditions (weaker winds, very stable/unstable, etc), for measurements at different altitudes, circle diameters, half opening angles, or anything else the authors think would be relevant to any user that would try to apply this method elsewhere.*
*Alternatively, instead of adding a discussion section the authors could expand their study to more time periods under different atmospheric conditions.*

The Referee points out that only five hours of data under a narrow set of conditions were used in the experimental results of this study. We agree that more data would, indeed, create a more solid foundation for our findings. The reasons for using such a limited data set are detailed in the answer to Referee #1.

Regarding the fact that there is a large disagreement between the modelled and the observed spectra, we added a second reference (Larsén et al., 2016). They show that spectra based on measurement data of complete years at Høvsøre also show large disagreement in the low wave number band when compared to model spectra. The desired investigation to understand the root cause of such differences lies clearly out of the scope of our study. In this context, the referee states that the model does not accurately represent real lidar measurements. But the strong disagreement between model and observed spectra is already found when the spectral tensor is compared solely to the sonic measurements. No actual filtering is part of this comparison, so one should not conclude from the disagreement that there is an insufficiency of the lidar models. The modelling of the different lidar data processing methods is mostly coherent with the experimental results, and the differences are analyzed in detail in the discussion of the results. We have no reason to assume that the conclusions

we draw would differ if a data set that is closer to the Mann spectral tensor would have been used. In order to make the reader aware of this problem, we emphasize in the text "that, due to the poor fit of the measured spectra of the horizontal wind components and the modeled spectra at low wave numbers, we can compare the relations between the different methods but not absolute values.".

We understand and share the wish for a more comprehensive study that covers several locations/seasons/times. And we regret that we cannot provide such in this paper. Our aim is to introduce a complete numerical description of a VAD scanning CW lidar for the first time and to extend it to two promising methods of modified data processing. Further, we want to give a detailed description of the cross-contamination effects that are briefly mentioned in previous publications but never fully described. The one set of experimental data we use supports this analysis and helps us to point out the limitations of the new methods and the models.

We agreed with the reviewer's suggestion to add a section on what we would expect from applying the methods to different conditions and measurement setups. This new section 6.4 discusses implications for varying measurement heights, cone angle, mean wind speeds and atmospheric stability conditions. This new section links our results to a possible wider adoption of cases.

*Specific comments:*
*a) P. 2 line 12: Somewhere around here it would be appropriate to reference Eberhard et al (1989) as one of the first studies where Doppler lidars were used to profile turbulence using VADs.*
  We added a reference to Eberhard et al. (1989) to the introduction: "The estimation of second order statistics of the turbulence in the wind by means of VAD scanning pulsed Doppler lidar was first demonstrated in Eberhard et al. (1989)."

*b) Figure 2: Parts of this figure (especially the subfigure showing the lidar beams) are extremely small and difficult to read. This should be improved, making the figure larger would help. I also suggest adding to the caption describing how the top plot visualizes the lidar beam positions.*
  We replaced the label "Lidar" by a laser symbol, clarified the caption and increased the overall size of the figure.

*c) P. 8 lines 5-13 & Fig. 3: This should be moved to later in the paper, perhaps Sect. 4. At this point, the reader has no context to understand the details of what is being shown as the model has not been described.*
  Thank you also for this suggestion. We agree that the origin and details of Fig. 3 are difficult to understand here and that an explanation would be beneficial to the reader at a later point. Unfortunately, we see dividing Sect. 2 into two pieces as the worse option. Also moving it as a whole after Sect. 4 would weaken the logical structure of the manuscript: namely, describe the problem in Sect. 2, sketch our approach to tackle it in Sect. 3 and model this approach in Sect. 4. Instead, we added information to contextualize the plots sufficiently so that they can be understood already here.

*d) Fig. 3 (and throughout): The units for power spectra in the atmospheric science community are generally m^-1, not rad m^-1. These units appear throughout the text, in the table, and figure.*
  We changed the units as suggested to m$^{-1}$.

*e) Eq. 10 & 11: Be consistent with these equations. Eq. 11 gives the entire variance for v while Eq. 10 only gives the variance contamination for u, but the subscript notation on the left-hand side for both indicates total variance. Also, Eq. 11 does not appear to be derived correctly (and is inconsistent with Eq. 10). Why is the ½ factor in the equation? The logic for how these Eqs are derived is not obvious (should be clarified), but should there also be a term for the covariance (u'w' and v'w' overbars) on the right-hand side?*
  Eq. 10 also gives the entire variance for u. Since we look at the situation at the resonance wave numbers like in the example in Fig. 2 the contribution of $u_{wind}$ to $u_{lidar}$ is zero. To better clarify

this for the reader, we have added "$_{res}$" to the subscript notation. A subscript "$_{unc}$" was added to Eq. 11 to emphasize that the equation is valid only when the inflow is uncorrelated between the two spatially separated points. This occurs at high wave numbers. We corrected the "tan" in eq. 11 to "cot". Thank you for this correction. We included a derivation of both equations so that the reader gets the logic behind them. The influence of the covariance terms averages out to zero.

*f) P. 9 lines 8-16, 20-21: This text should also be moved to Sect. 4 where the readers will have the appropriate context to understand the discussion here. The rest of the text after Eq. 11 and before Sect. 3 can be combined into one paragraph.*

Please find our answer in the response to c). We combined the last two paragraphs.

*g) Fig. 4: Nice figure. It could be improved if you added a reference vector for the mean wind to show how the field is advected. It would also be beneficial in the lower plot to highlight in a different color which pairs of measurements are used in the two-beam method.*

We are happy to get this suggestion and added the **U** vector to the figure. We also included a visualization of the beams used in 2-beam processing and mentioned this in the caption and referred to it in the text.

*h) Sect 4: Please make sure to explain and define all variables and notation. In particular, I could not find definitions for: e_1, Φ, and T. The notation of _z was also not described.*

We introduced the unit vector **e** and the meaning of the half cone opening angle Φ is now repeated in the text. We removed the index "$_z$" that was not giving additional clarity in its context.

*i) Sect 5.1: Add a sentence here cross-referencing table 1 to summarize the experiment. Please also expand this discussion. The time period is 5 hours, it is unlikely the wind speed was constant that entire time. How much did it vary? How much did the turbulence intensity vary? What was the stability of the boundary-layer? Given the location, I expect near-neutral stability, but it would be good to verify and quantify the stability. How were the resonance values in Table 1 determined? What was the precision of the lidar measurements?*

The min/max/std values of $U$ and $TI$ have been added and a reference to Table 1 is now given. We added the formulas for the resonance values given in Table 1 to section 2.5.1. The precision of the lidar is usually better than 1%. This information is added to the text. The stability of the boundary layer during the experiment was neutral, as shown below. $z/L$ is either derived from 10 m or 20 m with two different methods to calculate the momentum flux (2d and 3d) which give very similar results.

[Figure]

*j) P. 19 line 2: Please add a more throughout description of how these spectra are made. Is this an average of the individual 10-min spectra? Were outliers in the spectra removed in making this plot? How much did the actual spectra vary of the time period? The description is insufficient.*

Yes, the shown spectra are the averages of all individual spectra based on 10-min intervals. The resulting average spectra are then averaged within 30 wave number bins. Outlier detection in time domain was investigated but eventually not applied because its effect on this data set was negligible. This is already mentioned in subsection 5.2.

*k) P. 19 line 8: What is the 'target spectrum'? Is this simply the modeled spectrum assuming certain characteristics of the flow garnered from the sonic anemometer measurements?*

Very good point. The target spectrum is the unfiltered one-dimensional model spectrum based on the spectral tensor we chose to best-fit to the sonic anemometer measurements. We added the equation to derive the u spectrum from the spectral tensor to the end of the model description in section 4 and refer to it here for clarification.

*l) Fig. 5: These two plots can be combined as the axes are identical and much of the data overlaps. By combining the plots, the VAD/SMC and two-beam methods can also be compared. Also state in the caption what the vertical dashed lines indicate.*

We considered combining the subplots which, indeed, would give a better comparison between circle processing and 2-beam processing. Unfortunately, the plot appears overloaded then. We extended the caption by adding "… The grey vertical dashed lines represent the first and second resonance wave number.". The same applies to Fig. 7.

*m) P. 20 line 25: By this, do you mean that the frozen turbulence hypothesis is not completely valid as the wind field evolves in time as it advects through the measurement volume? If so, please clarify that here.*

Thank you for mentioning this point. The passage was unclear and we improved it as follows: "A possible explanation is that the fluctuations of the u- and especially the w-component in the real wind field are not perfectly correlated, i.e. the frozen turbulence hypothesis which the model assumes is slightly violated. The result is a small contribution of $w_{wind}$ on $u_{lidar}$ that appears to a higher extent in the VAD processed spectrum. The reason for the difference is that the correlation is closer to unity in the case of SMC processing."

*n) P. 20 line 34: Could this deviation also be caused by random errors in the lidar measurements resulting in a noise floor above the modeled value (related to last point in i) above)? Based on the model description in Sect. 4, measurements are modelled as precise (without any random error).*

To consider a random error in the lidar measurements as a cause for increased energy values in the measurements is reasonable. But for two reasons we do not believe it applies here. First, the effect should then be smaller in the whole circle processing methods than in the two beam approaches because random errors average out along the measurement circle. We don't see that in the measurements. Second, the effect of random erroneous line-of-sight measurements would lead to even stronger contamination of the w-spectra due to the higher sensitivity. This is also not represented in the results.

*o) Fig. 6: These two subplots can be combined (if kept, see comment p), as b) only contains one additional piece of information (red line) that could be easily overlaid on a) for comparison.*

We combined both subplots and swapped the color of the "2-beam model" line to cyan in all plots for better distinction.

*p) Sect. 6.2.2: This section and Fig. 6 b can be removed. If this method is not even applicable to real CW lidar measurements due to the ambiguity around a 0 Doppler velocity, why even present it as a method?*

We added that "We included the model behavior of 2-beam processing for the sake of completeness and to show that the availability of reliable measurement data from the east and west beams would be of hardly any use."

*q) P. 25 line 20: Recommend changing the term from 'very good' to 'reasonable'. There are still non-trivial differences between the modeled and observed spectra in this reviewer's opinion.*

We now describe the forecasting as 'reasonable'.

*Editorial corrections*
*a) P. 1 Line 19: Add hyphen to Velocity-azimuth display.*
*b) P. 6 line 30: Reword to: 'Turbulence with a length scale below...'*
*c) P. 7 line 1: Remove 'to sense them'*
*d) p.7 line 10: Remove 'or cross talk'*
*e) P. 13 line 4: 'is' should be 'are'*
*f) Eq. 23: Should be 'sin' instead of 'sinc'.*
*g) P. 18 line 2: Remove 'used'*

We thank the reviewer for mentioning these errors which we have corrected. "sinc" is not a mistake but the abbreviation for the cardinal sine function that must be used. We now give its definition in the text. Mathematics is singular.

*References:*

[revised manuscript text omitted]

---

## Author Response (AR2)

We thank both referees a lot for their work put into reviewing our manuscript. Their feedback made it possible to improve the text significantly during the revision phase.

Referee #1 suggested to include the reasoning for using a very limited data set in the study into the manuscript. We added this information to the extended discussion.

Based on the final report from Referee #2 we checked the values of the wave numbers $k$ carefully to make sure they are correct.

[revised manuscript text omitted]